# Varying relationships between experienced income segregation and travel behaviour across neighbourhood social and urban contexts

Yuxuan Zhou ◉ & Yi Lu ◉ ✉

Income segregation is a barrier to social inclusivity and equity. It is affected by individuals' travel behaviour and socioeconomic status and may be intensified by localised living models emphasising conducting daily activities within immediate neighbourhoods. However, how the relationship between experienced income segregation and travel behaviour varies across neighbourhood social and urban contexts remains unclear. Here, we quantify experienced income segregation using a dataset of 1.2 billion mobility records from the contiguous United States and examine its relationships with travel distance and diversity across neighbourhoods with different social and urban contexts. We find that longer travel distances and more diverse destinations are associated with less experienced segregation for least affluent neighbourhoods, especially in less urbanised areas. Our findings underscore the need for urban planning and transport interventions to increase mobility and social integration opportunities for residents from socially disadvantaged neighbourhoods. They also highlight the potential adverse social consequences of localised living models.

In the United States and many other countries, socioeconomic segregation, which is the extent to which populations with different socioeconomic status (SES) separate with each other, is a persistent and worsening social issue[1,2]. It imposes a substantial barrier to inclusive and equitable urban development, contributing to a wide range of social issues such as health inequalities[3,4], crime[5,6], and unequal access to essential resources[7,8]. There is well-established evidence showing that income segregation in residential contexts has adverse effects on social inequality and health[9,10]. However, individuals also experience income segregation across a range of activity sites beyond their residential areas. Recent studies suggest that experienced income segregation (hereafter 'experienced segregation') in these non-residential settings is generally lower than that observed in residential areas[11]. This multicontextual perspective provides an important theoretical

framework to understand how income segregation is shaped by residents' daily travel and activities.

Travel behaviours affect where a person goes and whom they encounter in their daily lives. It is intuitive that people who travel further from their neighbourhoods and visit more diverse destinations are more likely to meet people with different classes than those who stay close to home, thereby reducing their experienced segregation[12–14]. However, to achieve sustainable urban development, many urban planners and policymakers now advocate for localised living models, such as the 15-min city, compact city, or new urbanism models, that encourage residents to access essential amenities and meet their daily needs in close proximity to their homes[15–17]. It is argued that such models not only minimise transportation energy consumption but also create strong community bonds, as residents live, work,

---

Department of Architecture and Civil Engineering, City University of Hong Kong, Hong Kong Special Administrative Region, Hong Kong, China.
✉e-mail: yilu24@cityu.edu.hk

and socialise in close proximity to each other[18]. However, a high degree of localised living could reduce residents' travel distance and travel diversity, thereby limiting interactions across social groups and potentially reinforcing class divides, particularly for socially disadvantaged residents with constrained travel options and limited mobility[15,19,20]. Moreover, activity sites within localised living circles may be predominantly occupied by socially homogeneous populations from surrounding neighbourhoods, particularly in areas with high residential segregation, thereby intensifying segregation by limiting opportunities to access activity sites beyond the residential environment[21]. Hence, addressing the trade-off between localised living and reduced experienced segregation requires a nuanced understanding of the role of travel behaviour in shaping experienced segregation.

Studies have documented clear disparities in travel behaviour across social groups. Socially advantaged individuals have greater flexibility in choosing where to live, work, and socialise, potentially leading to more diverse travel behaviours. In contrast, socially deprived individuals, constrained by housing affordability and low incomes, tend to have limited travel options and less efficient commutes[19,22–26]. These individuals, typically residing in socioeconomically deprived neighbourhoods, may experience increased income segregation risks within localised living models. Conversely, access to a broader range of activity sites may promote greater interaction across socioeconomic boundaries for socially disadvantaged residents, with travel behaviour playing a critical role in facilitating these benefits[12,27,28]. Nevertheless, the link between travel behaviour and experienced segregation for different social groups remains largely unexplored. Conducting a nationwide analysis of how travel behaviour influences experienced segregation across social groups remains challenging[12,29–31], primarily due to the difficulty of linking income data with mobility records and measuring real-world exposure to others within activity sites. Moreover, most research on segregation has focused on highly urbanised settings (i.e., metropolitan areas) and has rarely examined less urbanised contexts, such as micropolitan areas, small towns, and rural areas[11,12,20,28,29,32,33], where residents may exhibit distinct social backgrounds and travel behaviours. To develop a more comprehensive understanding of how experienced segregation relates to travel behaviour, it is essential to extend the spatial scope beyond metropolitan areas and consider diverse urban contexts.

Here, we leverage a large-scale, neighbourhood-level mobility dataset from the contiguous United States to quantify experienced segregation in activity sites and compare it with residential segregation across neighbourhoods with different social and urban contexts. To examine the relationships between experienced segregation and travel behaviour, we conduct two sets of analyses, incorporating both intra-neighbourhood and inter-neighbourhood comparisons. In the first analysis, we decompose experienced segregation for each neighbourhood using the travel distances recorded for each mobility event and compare the experienced income segregation of trips with different travel distances. In the second analysis, we perform regression analyses on the correlation between travel distance, travel diversity and experienced segregation, separately for neighbourhoods within each urbanicity level. This inter-group comparison enables a systematic assessment of how neighbourhood travel behaviour relates to overall experienced segregation. Overall, this study makes two contributions. First, we establish a nationwide picture of experienced income segregation in activity sites across neighbourhoods with different social and urbanicity contexts. We show that individuals from neighbourhoods in the highest income quartile in metropolitan areas experience higher levels of experienced segregation, while those from neighbourhoods in quartiles 1 and 2 (i.e., low-income neighbourhoods) in less urbanised areas, including micropolitan areas, small towns, and rural areas, also face heightened segregation. Notably, residents from low-income and majority-POC (people of colour) neighbourhoods experience a greater reduction in segregation when comparing experienced segregation to residential segregation. Second, we offer a nuanced understanding of how travel behaviour, specifically travel distance and travel diversity, relates to experienced segregation. Our study reveals that individuals from low-income neighbourhoods, especially in small towns and rural areas, benefit more from reduced experienced segregation when undertaking longer and more diverse trips. Furthermore, those from majority-POC neighbourhoods and those living in areas with high residential segregation particularly benefit from longer travel distances. Such findings have implications for policymakers who favour localised living models, because reducing travel distances may alleviate travel-associated emissions, but at the expense of limiting economic upward mobility and social interaction for residents of low-income neighbourhoods. These findings underscore the importance of enhancing transport accessibility and affordability for the least affluent neighbourhoods, especially those with high residential segregation, as a part of an inclusive and equitable society, amid the rising popularity of localised living theory.

## Results

### Overall patterns of experienced segregation

We estimated experienced segregation using a large-scale human mobility dataset comprising 1,226,450,275 origin–destination (OD) flows from census block groups (CBGs) to activity sites, defined by Points of Interest (POIs). Even though CBGs are the next level above census blocks, income data from the American Community Survey (ACS) are not available at the block level; CBGs are thus the smallest geographic units for which mobility and income data are available. We conducted our analyses at the CBG level, which we refer to as neighbourhood in this study. To quantify experienced segregation for residents of each neighbourhood, we first constructed an income dissimilarity matrix using the ranked median household income of each neighbourhood, following the concept of 'income distance' developed by Xu et al.[34]. The income dissimilarity from one focal neighbourhood to another is defined as the proportion of all other neighbourhoods whose income distance to the focal neighbourhood is smaller than that between the two (see 'Methods' for details). This matrix captures the income dissimilarity between all pairs of neighbourhoods. Experienced income dissimilarity of individuals from a neighbourhood in each activity site was calculated as the weighted average income dissimilarity between their neighbourhood and the neighbourhoods of other individuals present in the same site, with weights based on the number of individuals from each other neighbourhood. The overall experienced segregation of each neighbourhood was calculated as one minus the weighted average of experienced income dissimilarity across all activity sites visited by its residents, with weights based on the number of individuals from the given neighbourhood present in each activity site. Values of experienced segregation range from 0 to 1, with higher values indicating greater levels of segregation. To determine the heterogeneity of experienced segregation across urbanicity, we categorised neighbourhoods into four urbanicity levels based on the rural–urban commuting area (RUCA) Codes: metropolitan areas, micropolitan areas, small towns, and rural areas.

Overall, the median experienced segregation across all neighbourhoods is 0.594, with variations across different types of destinations (Supplementary Fig. 1). Spatially, areas with relatively high segregation levels are predominantly small towns and rural areas, whereas areas with relatively low segregation levels are predominantly found in metropolitan and micropolitan areas (Fig. 1a). Three metropolitan statistical areas (MSA) with distinct average value of experienced segregation further show that central areas exhibit relatively low experienced segregation, whereas fringe areas display high experienced segregation (Fig. 1b, c). A gradient decline in experienced

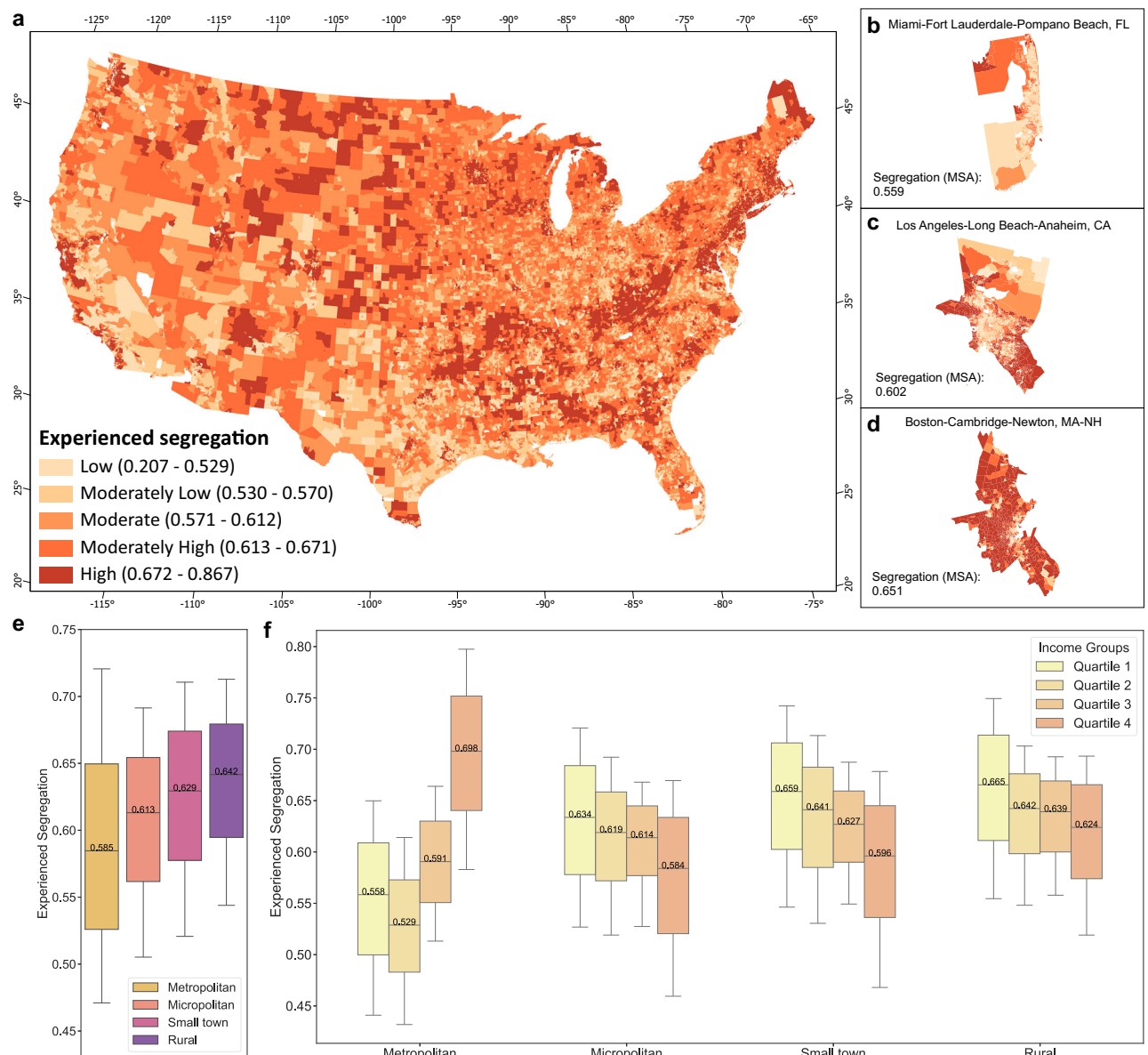

**Fig. 1 | Patterns of experienced segregation. a** Spatial patterns of experienced segregation in the contiguous United States (presented here at the census tract level rather than the census block group (CBG) level for the purpose of data visualisation). We selected and zoomed in on three representative metropolitan statistical areas (MSAs) with distinct levels of experienced segregation: **b** Miami−Fort Lauderdale−Pompano Beach, Florida (average segregation: 0.559), **c** Los Angeles−Long Beach−Anaheim, California (average segregation: 0.602), and **d** Boston−Cambridge−Newton, Massachusetts−New Hampshire (average segregation: 0.651). The experienced segregation was categorised into quintiles, with quintiles 1–5 corresponding to low, moderate-low, moderate, moderate-high, and high levels, respectively. **e** Distribution of overall experienced segregation at the neighbourhood level across different urbanicity levels. **f** Distribution of overall experienced segregation across various income quartiles in distinct urbanicity levels, with income quartiles defined separately for each urbanicity level. In **e**, **f** $n = 167,780, 20,580, 10,847, 8,683$ neighbourhoods in metropolitan areas, micropolitan areas, small towns, and rural areas, respectively. The box plots in **e**, **f** present the 10th, 25th, 50th, 75th, and 90th percentiles.

segregation is observed with increasing urbanicity, with median values of 0.585, 0.613, 0.629, and 0.641 for metropolitan, micropolitan, small town, and rural areas, respectively (Fig. 1e). Additionally, income disparities in experienced segregation are evident across urbanicity levels (Fig. 1f), with individuals from low-income neighbourhoods in metropolitan areas (quartile 1 and quartile 2) experiencing less income segregation than those from high-income neighbourhoods (quartile 3 and quartile 4). Nevertheless, this pattern reverse in less urbanised areas, where individuals from lower-income neighbourhoods experience higher income segregation than those from higher-income neighbourhoods in micropolitan areas, small towns, and rural areas. This indicates that highly urbanised cities provide greater opportunities for

individuals from lower-income neighbourhoods to encounter people from diverse class. These patterns are also evident across different activity sites (Supplementary Fig. 2). In addition to income disparities, we also examined racial disparities in experienced segregation. In metropolitan areas, majority-POC neighbourhoods, particularly those that are also low-income (neighbourhoods in income quartiles 1 and 2), tend to exhibit relatively low levels of experienced segregation compared to majority-White neighbourhoods, whereas these same groups experience relatively higher income segregation in less urbanised areas, including micropolitan areas, small towns, and rural areas (Supplementary Fig. 3 and Supplementary Fig. 4). The relatively high accessibility to public transit and more transportation options in

metropolitan areas enables residents to travel more easily to various destinations and interact with people from diverse socioeconomic backgrounds compared to those in less urbanised regions[35]. Majority-POC neighbourhoods, particularly those with low income, are likely to benefit more from these enhanced opportunities for mobility and social interaction in metropolitan settings. In contrast, in less urbanised areas, limited accessibility to amenities constrains residents of low-income, majority-POC neighbourhoods from visiting areas frequented by more diverse populations. Meanwhile, residents of high-income neighbourhoods, regardless of racial composition, generally have greater access to resources and mobility, allowing them more opportunities to visit destinations with heterogeneous visitors.

## Comparison of experienced and residential segregation

To further examine the relationship between experienced and residential segregation, we calculated residential segregation for each neighbourhood. Similar to experienced segregation, residential segregation was defined as one minus the weighted average income dissimilarity between a given neighbourhood and its adjacent neighbours, with weights based on the population size of each neighbour. Neighbourhood adjacency was determined using the queen contiguity approach, which has been widely used in spatial analysis. Unlike experienced segregation, residential segregation is relatively higher in both metropolitan and rural areas. In metropolitan areas, neighbourhoods in both the lowest and highest income quartiles exhibit elevated levels of residential segregation, which we term 'self-segregation'. In contrast, a clear income gradient in residential segregation is observed in less urbanised areas, with segregation levels decreasing as income increases (Supplementary Fig. 5). The distribution of experienced segregation closely mirrors that of residential segregation across urbanicity levels (Spearman correlation = 0.64 in metropolitan areas, 0.61 in micropolitan areas, 0.72 in small towns, and 0.70 in rural areas; $p < 0.001$), with experienced segregation generally lower. Specifically, more than 70% of neighbourhoods exhibit lower experienced segregation than residential segregation across all urbanicity levels: 78.2% in metropolitan areas, 72.4% in micropolitan areas, 71.7% in small towns, and 78.5% in rural areas (Fig. 2a). Among these neighbourhoods, experienced segregation is lower than residential segregation by a median of 17.48% in metropolitan areas, representing the largest disparity, followed by 13.85% in micropolitan areas, 12.88% in rural areas, and 11.93% in small towns (Fig. 2b).

Regarding income disparities, neighbourhoods in the lowest income quartile in metropolitan areas exhibit the greatest residential-experienced segregation disparity, with experienced segregation exhibiting lower than residential segregation by a median of 24.07%, followed by 18.19% in quartile 2, 15.06% in quartile 3, and 14.52% in the highest-income neighbourhoods. A similar pattern is observed in micropolitan areas and small towns, where neighbourhoods in the lowest income quartile also experience the largest reduction in segregation, while this trend is less clear in rural areas (Fig. 2c). To facilitate comparison, we grouped neighbourhoods in income quartiles 1 and 2 as the low-income group, and those in quartiles 3 and 4 as the high-income group. For racial disparities, majority-POC neighbourhoods, particularly those with low income, present greater residential-experienced segregation disparity across urbanicity levels (Supplementary Fig. 6 and Supplementary Fig. 7). This indicates that socially disadvantaged neighbourhoods (i.e., majority-POC neighbourhoods with low income) experience greater reduced income segregation in activity sites compared to residential areas, especially in metropolitan areas. The result shows that low-income neighbourhoods with relatively high residential segregation in highly urbanised areas experience the greatest disparity between residential and experienced segregation, whereas high-income neighbourhoods with lower residential segregation across all urbanicity levels show the smallest disparity (Fig. 2d).

## Experienced segregation by trips of varying travel distances

To compare experienced segregation by trips with different travel distances, we first classified all trips as either inside-neighbourhood or outside-neighbourhood for each neighbourhood, assuming that trips beyond the home neighbourhood involve greater travel distances. The results show that experienced segregation is consistently higher for inside-neighbourhood trips than for trips outside the home neighbourhoods across income groups in all four urbanicity levels (Supplementary Fig. 8). Particularly, both high- and low-income neighbourhoods with high residential segregation demonstrate greater reductions in experienced segregation during trips outside their own neighbourhoods relative to trips inside them (Supplementary Fig. 9). Moreover, majority-POC neighbourhoods with low income present the greatest reduction in outside-neighbourhood trips related to inside-neighbourhood trips (Supplementary Fig. 10 and Supplementary Fig. 11). Notably, this disparity becomes more pronounced as urbanicity decreases.

We further classified all trips into quartiles based on trip distance within each urbanicity level, and the experienced segregation was measured separately for each distance quartile. We observed a gradient decrease in experienced segregation from quartile 1 (shortest trips) to quartile 4 (longest trips) of trip distance across different urbanicity levels, with the most pronounced disparities evident in rural areas (Fig. 3a). Specifically, segregation is reduced by 11.59% (95% confidence interval (CI): 11.52–11.67%), 17.20% (95% CI: 16.93–17.47%), 19.40% (95% CI: 18.87–19.94%), and 21.84% (95% CI: 21.50–22.17%) for the longest trips (quartile 4) compared with short trips (quartile 4) in metropolitan areas, micropolitan areas, small towns, and rural areas, respectively. Moreover, strong SES disparity is observed, with lower-income people being more sensitive to trip distance (Fig. 3b–e). More specifically, the most deprived neighbourhoods in metropolitan areas, micropolitan areas, small towns, and rural areas experience reductions in segregation of 18.65% (95% CI: 18.50–18.80%), 21.80% (95% CI: 21.51–22.08%), 24.58% (95% CI: 24.21–24.95%), and 27.90% (95% CI: 27.52–28.29%), respectively, between trips with the longest distance (quartile 4) and trips with the shortest distance (quartile 1). In comparison, the most affluent neighbourhoods in metropolitan areas, micropolitan areas, small towns, and rural areas only experience reductions of 11.22% (95% CI: 11.12–11.34%), 6.02% (95% CI: 5.18–6.86%), 7.65% (95% CI: 6.72–8.59%), and 9.01% (95% CI: 8.15–9.87%), respectively. In addition, neighbourhoods with high residential segregation, especially those with low-income, show greater reduction in experienced segregation during the longest trips compared to the shortest ones (Supplementary Fig. 12). Similarly, greater reductions in experienced segregation with increasing travel distance are also observed among majority-POC neighbourhoods, especially those that are low-income (Supplementary Fig. 13 and Supplementary Fig. 14).

We further compared experienced segregation during trips of varying travel distances to residential segregation and observed a clear distance gradient: longer trips to activity sites are associated with greater disparity between experienced segregation relative to residential segregation. This gradient is particularly steep in rural areas, where the reduction in experienced segregation compared to residential segregation during the longest trips is 88.14% greater than that observed during the shortest trips. In contrast, in metropolitan areas, the corresponding difference is 53.40% (Fig. 4a). These results suggest that longer-distance activities more effectively mitigate residential segregation than shorter-distance activities, particularly in less urbanised settings. With respect to income disparities, low-income neighbourhoods, especially those with higher residential segregation, exhibit substantially greater residential-experienced segregation disparities during the longest trips compared to the shortest, most notably in less urbanised areas (Fig. 4b–e, Supplementary Fig. 15). Moreover, majority-POC neighbourhoods, particularly those that are also low-income, experience greater residential-experienced

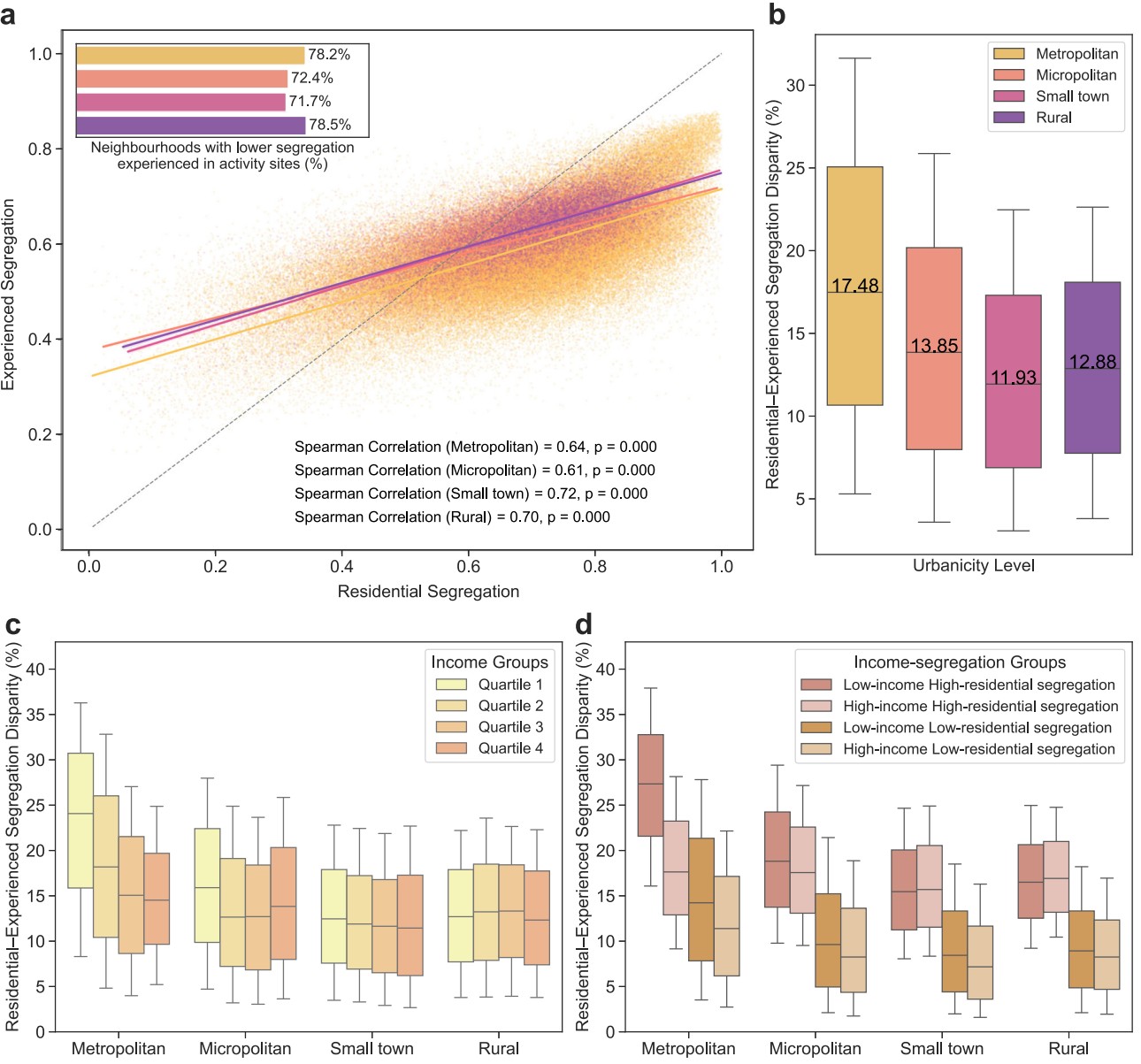

**Fig. 2 | Relationships between experienced segregation and residential segregation.** **a** Scatter plots and Spearman correlations between residential and experienced segregation across urbanicity levels ($n = 167,780, 20,580, 10,847$, and 8683 neighbourhoods in metropolitan areas, micropolitan areas, small towns, and rural areas), using two-sided significance tests. The grey dashed line represents the line of equality ($y = x$), indicating equal levels of experienced and residential segregation. The bar chart in the upper left corner shows the proportion of neighbourhoods with lower experienced segregation than residential segregation. **b** Disparities between residential and experienced segregation across urbanicity levels, calculated as (residential segregation − experienced segregation)/residential segregation. Only neighbourhoods where experienced segregation is lower

than residential segregation were included in the calculation. **c** Residential-experienced segregation disparities by income groups and urbanicity levels. **d** Residential-experienced segregation disparities by income and residential segregation levels. In each urbanicity level, neighbourhoods in income quartiles 1 and 2 were classified as the low-income group, and those in quartiles 3 and 4 as the high-income group. Within each income group of each urbanicity level, neighbourhoods were further divided into low and high residential segregation groups based on the group-specific median. In **b**–**d**, $n = 131,278, 14,905, 7776$, and 6820 neighbourhoods in metropolitan areas, micropolitan areas, small towns, and rural areas, respectively. The box plots in **b**–**d** present the 10th, 25th, 50th, 75th, and 90th percentiles.

segregation disparities during longer trips than shorter ones (Supplementary Fig. 16 and Supplementary Fig. 17). These findings indicate that socially disadvantaged neighbourhoods benefit more from mitigating residential segregation by experiencing reduced experienced segregation in longer-distance trips.

To further illustrate how travel distance shapes experienced segregation, we calculated, for each type of activity site, the average segregation experienced by individuals from all neighbourhoods visiting that space type, along with the corresponding average travel distance. We find consistent negative correlations between the

experienced segregation associated with specific types of activity sites and the average travel distance required to reach them across urbanicity levels. For example, in metropolitan areas, trips to schools are substantially shorter than trips to hotels, by 85.09% (95% CI: 84.83–85.35), 89.20% (95% CI: 89.07–89.33), 90.80% (95% CI: 90.68–90.92), and 91.84% (95% CI: 91.71–91.97) for income quartiles 1 through 4, respectively. However, individuals from these neighbourhoods experience 9.93% (95% CI: 9.79–10.06), 4.96% (95% CI: 4.83–5.10), 5.26% (95% CI: 5.17–5.35), and 10.33% (95% CI: 10.25–10.42) greater segregation in schools compared to hotels (Fig. 5a). The

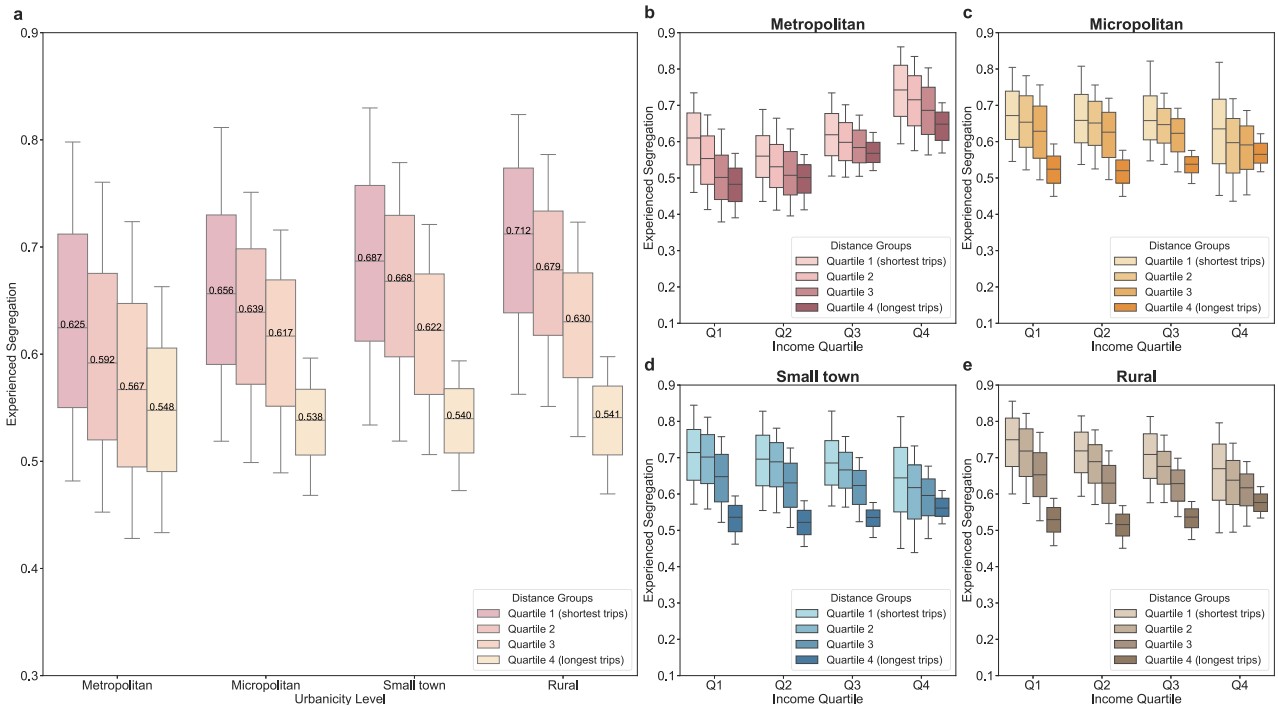

**Fig. 3 | Patterns of experienced segregation decomposed by travel distance across urbanicity levels and income groups. a** Decomposition of experienced segregation by trip distance quartiles (from quartiles 1–4) across urbanicity levels (*n* = 167,780, 20,580, 10,847, and 8683 neighbourhoods in metropolitan areas, micropolitan areas, small towns, and rural areas, respectively). **b**–**e** Evident income disparities in experienced segregation by travel distance quartiles across urbanicity levels. In **b**–**e**, Q1, Q2, Q3, and Q4 denote income quartiles 1–4, respectively. For

metropolitan areas, *n* = 41,977, 41,913, 41,946, and 41,944 neighbourhoods in income quartiles 1–4. For micropolitan areas, *n* = 5146, 5144, 5145, and 5145 neighbourhoods in income quartiles 1–4. For small towns, *n* = 2713, 2711, 2711, and 2712 neighbourhoods in income quartiles 1–4. For rural areas, *n* = 2171, 2177, 2166, and 2169 neighbourhoods in income quartiles 1–4. The box plots in **a**–**e** present the 10th, 25th, 50th, 75th, and 90th percentiles.

associations are particularly pronounced for low-income neighbourhoods in less urbanised areas (Fig. 5b, c), especially those with a majority population of colour and a high level of residential segregation (Supplementary Figs. 18–20). These findings suggest that activity sites requiring shorter travel distances (e.g., schools) may disproportionately serve socially homogeneous local populations, thereby reinforcing experienced segregation[11,12]. In contrast, non-essential activity sites that require longer travel distances, such as hotels, may promote greater social integration, particularly for socially disadvantaged groups.

### Neighbourhood travel behaviour and experienced segregation

To examine the associations between travel distance and experienced segregation at the neighbourhood level, we categorised neighbourhoods into quintiles based on visitor-weighted average travel distances for each income group across different urbanicity levels. As shown in Fig. 6a, there is a significant decrease in experienced segregation with increasing travel distance for neighbourhoods in the lowest income quartile (quartile 1) in metropolitan areas. Specifically, neighbourhoods with the highest quintile of travel distance experience 9.15% less segregation than those with the lowest quintile. In contrast, there is an increase in average experienced segregation with increasing average travel distance for higher-income quartiles (quartile 3 and quartile 4). In less urbanised areas, for neighbourhoods in income quartiles 1 and 2, a significant decrease in average experienced segregation is observed between those with longer travel distances and those with shorter average distances (Fig. 6b–d). In contrast, for neighbourhoods in the highest income quartile across micropolitan areas, small towns, and rural areas, the segregation increased with travel distance. These findings were further confirmed in the regression analyses. In metropolitan areas, the coefficient for travel distance in income quartiles 1

and 2 are −0.219 (*p* < 0.001) and −0.011 (*p* = 0.021) after controlling covariates, whereas the coefficients in income quartiles 3 and 4 are positive (Fig. 6e). In comparison, neighbourhoods in the 1st to 3rd income quartiles exhibit negative associations between travel distance and experienced segregation in micropolitan areas, small towns, and rural regions, whereas neighbourhoods in quartile 4 demonstrate a consistent positive association (Fig. 6f–h). Additionally, the negative associations between experienced segregation and travel distance among neighbourhoods in the lower income quartiles are stronger in less urbanised areas (e.g., rural areas). The results of the full regression models are presented in Supplementary Table 4.

Similar to the analysis of travel distance, we categorised trips into quintiles by travel diversity for various income groups across different urbanicity levels. Travel diversity is defined as the extent to which residents from a neighbourhood equally travel to different types of activity sites and is measured for each neighbourhood by the entropy score of trips to different types of activity sites (see 'Methods' for details). The results show that there is a significant decrease in experienced segregation with increasing travel diversity for income quartiles 1 and 2 in metropolitan areas, whereas the pattern is reversed for income quartiles 3 and 4 (Fig. 7a). This pattern is consistent and more pronounced in less urbanised areas (Fig. 7b–d). Particularly, for neighbourhoods in the lowest income quartile 1, those within the highest travel diversity quintile experience 7.81%, 10.24%, and 10.26% less segregation in micropolitan areas, small towns, and rural areas, respectively, than those in the lowest travel diversity quintile. The regression analyses confirm these findings. In metropolitan areas, the coefficients for travel diversity are −0.017 (*p* < 0.001) and −0.142 (*p* < 0.001) for income quartiles 1 and 2, respectively, and 0.088 (*p* < 0.001) and 0.116 (*p* < 0.001) for quartiles 3 and 4, respectively (Fig. 7e), indicating a negative association between travel diversity and

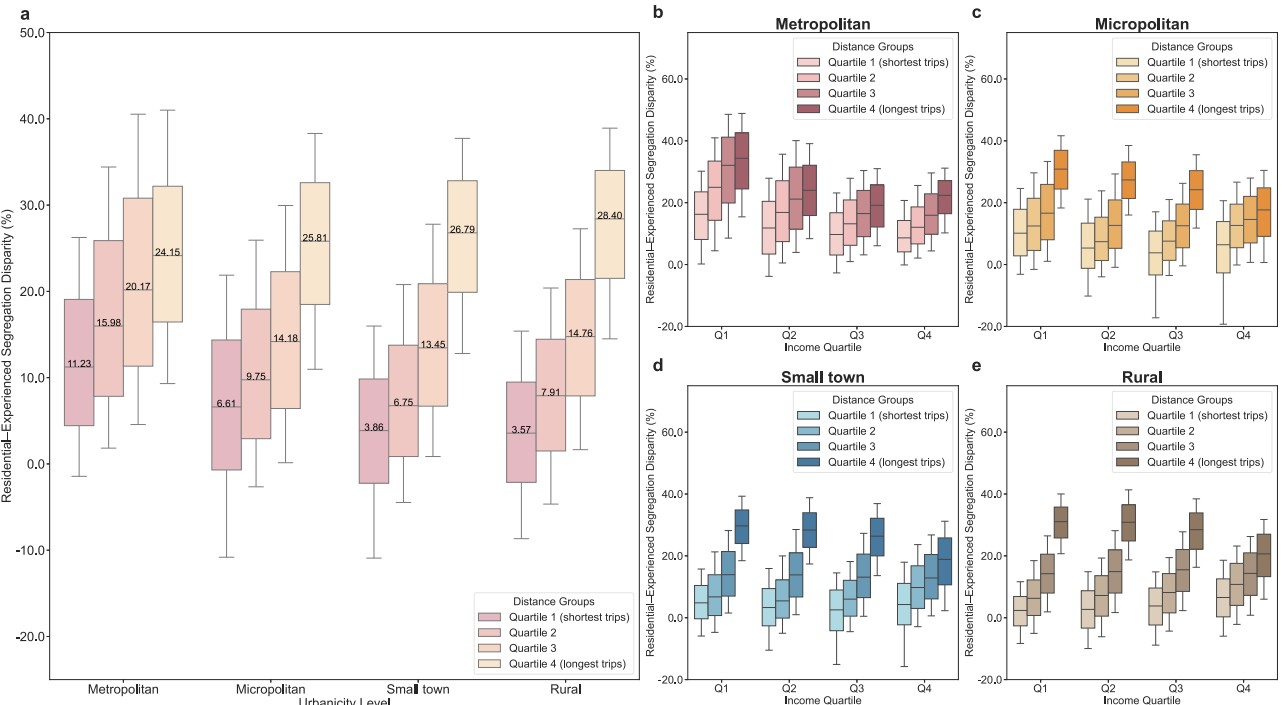

**Fig. 4 | Disparities between residential segregation and experienced segregation decomposed by travel distance across urbanicity levels and income groups. a** Disparities between residential and experienced segregation decomposed by travel distance quartiles across urbanicity levels. Only neighbourhoods where overall experienced segregation is lower than residential segregation were included in the calculation ($n = 131,278$, 14,905, 7776, and 6820 neighbourhoods in metropolitan areas, micropolitan areas, small towns, and rural areas, respectively). **b–e** Residential-experienced segregation disparities by travel distance quartiles across income groups and urbanicity levels. In **b–e**, Q1, Q2, Q3, and Q4 denote income quartiles 1–4, respectively. For metropolitan areas, $n = 32,820$, 32,821, 32,824, and 32,813 neighbourhoods in income quartiles 1–4. For micropolitan areas, $n = 3727$, 3728, 3732, and 3718 neighbourhoods in income quartiles 1–4. For small towns, $n = 1944$, 1944, 1945, and 1943 neighbourhoods in income quartiles 1–4. For rural areas, $n = 1705$, 1706, 1704, and 1705 neighbourhoods in income quartiles 1–4. The box plots in **a–e** present the 10th, 25th, 50th, 75th, and 90th percentiles.

experienced segregation for neighbourhoods in quartiles 1 and 2, but a positive association for those in quartiles 3 and 4. These patterns are more pronounced in less urbanised areas (Fig. 7f–h). For neighbourhoods income quartile 1 in micropolitan areas, small towns, and rural areas, the coefficients are $-0.224$ ($p < 0.001$), $-0.317$ ($p < 0.001$), and $-0.359$ ($p < 0.001$), respectively, whereas the coefficients are $-0.266$ ($p < 0.001$), $-0.282$ ($p < 0.001$), and $-0.157$ ($p < 0.001$), respectively, for neighbourhoods in income quartile 2. These indicate that for lower-income neighbourhoods in less urbanised areas, those with greater travel diversity exhibit greater reduced experienced segregation. The results of the full regression models are presented in Supplementary Table 5.

To further investigate heterogeneity in the observed associations, we conducted subsample analyses exclusively for low-income neighbourhoods (i.e., those in income quartiles 1 and 2), examining whether the negative relationships between travel distance, travel diversity, and experienced segregation differ across neighbourhoods with varying levels of residential segregation, racial composition, and access to public transit. Among low-income neighbourhoods in metropolitan areas, small towns and rural areas, those with higher residential segregation present stronger negative relationships between experienced segregation and average travel distance, indicating that these neighbourhoods with longer average travel distances tend to experience greater reduction in segregation (Supplementary Fig. 21). However, neighbourhoods with high residential segregation do not show similarly greater reduced segregation with increasing travel diversity (Supplementary Fig. 22). Regarding racial disparities, low-income, majority-POC neighbourhoods exhibit greater reductions in experienced segregation with increasing travel distances compared to their majority-White counterparts in metropolitan areas and small towns,

while this pattern is not significant among micropolitan and rural neighbourhoods (Supplementary Fig. 23). In contrast, low-income majority-White neighbourhoods show greater reductions in experienced segregation with increasing travel diversity across metropolitan, micropolitan, and small-town areas (Supplementary Fig. 24). For public transits, while low-income neighbourhoods lacking public transit within a 15-minute walking distance show higher experienced segregation, only those in rural areas experience significantly greater segregation reduction with increasing travel distance (Supplementary Fig. 25). Across all urbanicity levels, low-income neighbourhoods without nearby public transit consistently exhibit greater reductions in experienced segregation as travel diversity increases (Supplementary Fig. 26), suggesting that neighbourhoods without accessible public transit are more sensitive to the relationship travel diversity and experienced segregation. These results suggest that access to nearby public transit plays a key role in supporting longer travel distances that mitigate experienced segregation in rural low-income neighbourhoods, and in promoting travel diversity and reducing experienced segregation in low-income neighbourhoods across urbanicity levels.

## Discussion
Travel behaviour plays an important role in shaping where an individual goes and whom he or she meets[26,36] and contributes to the income segregation experienced by individuals during daily life, which has been shown to substantially differ from residential segregation[11,12,28,37,38]. However, the nuanced understanding of the relationships between travel behaviour and experienced segregation under different social and urban contexts remains unexplored. To address these crucial gaps, this study uses a nationwide human mobility dataset to conduct a comprehensive assessment of

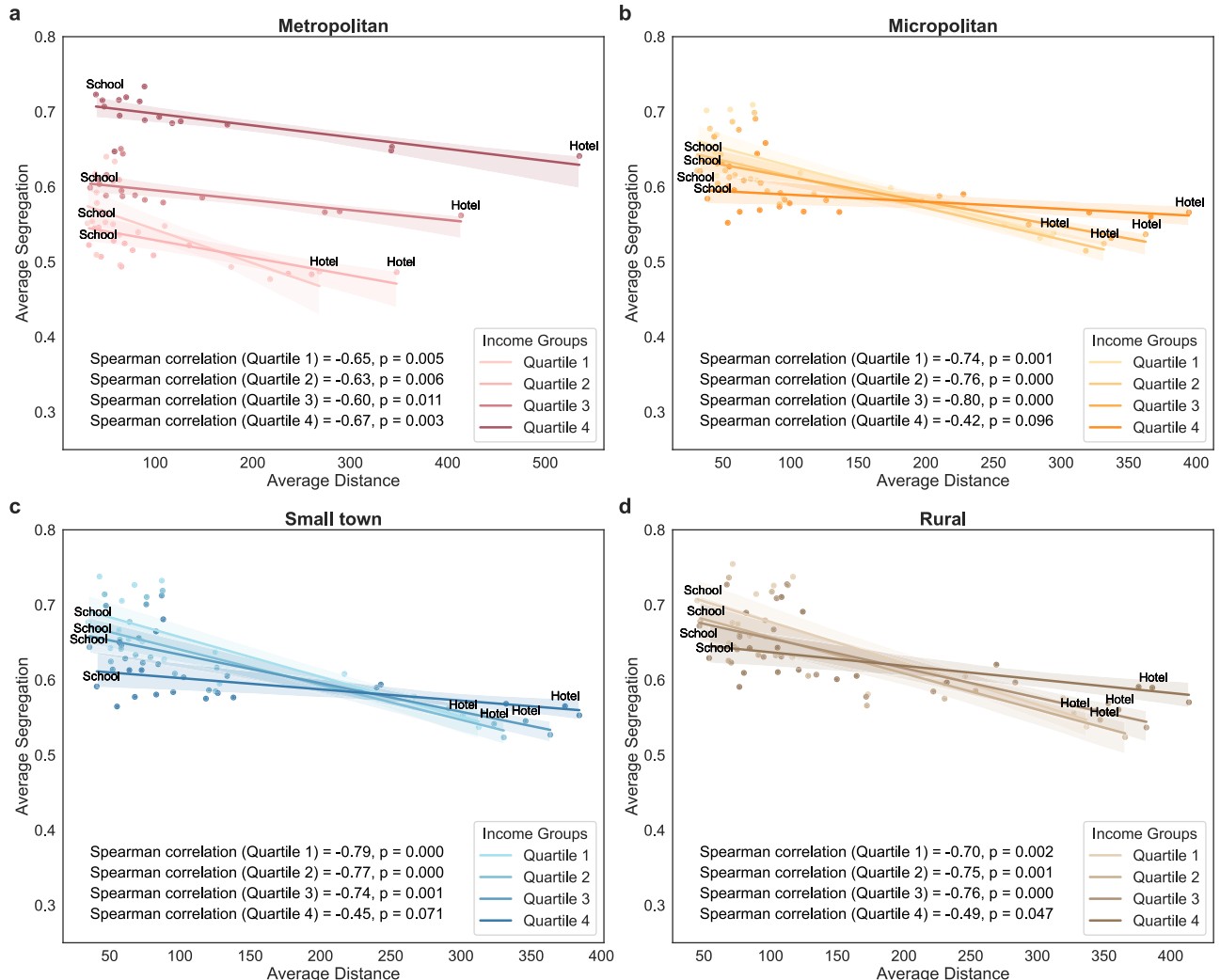

**Fig. 5 | Associations between average experienced segregation and average travel distance at the level of activity sites.** Spearman correlations between experienced segregation and travel distance for activity sites in **a** metropolitan areas, **b** micropolitan areas, **c** small towns, and **d** rural areas. Statistical significance was assessed using two-sided tests. Each dot represents a specific type of activity site within an income quartile, with $n = 16$ types. Lines represent linear associations for each income quartile, with shaded bands indicating 95% confidence intervals.

experienced segregation and its relationships with travel behaviour across social groups and varying urbanicity levels in the contiguous United States. Our study highlights the role of long and diverse trips in reducing experienced segregation for residents of the socially deprived neighbourhoods, which may seem counterintuitive amid the popularity of localised living models[18]. Localised living models aim to enhance residents' access to essential amenities within their immediate communities, thereby reducing the need for long-distance travel. While such models aim to create environmentally sustainable cities, they may unintentionally increase income segregation by shortening travel distances. Our findings provide critical insights in three key respects.

First, our study highlights how experienced segregation differs from residential segregation across social groups and urbanicity levels. While experienced segregation is strongly correlated with residential segregation, it tends to be lower, particularly in metropolitan areas. While a clear gradient of decreasing experienced segregation is observed with increasing urbanicity, residential segregation remains high in metropolitan settings, where self-segregation is evident among both the lowest- and highest-income neighbourhoods. Prior research has documented the prevalence of income inequality and self-segregation under residential contexts in highly urbanised areas,

contributing to increased levels of residential income segregation in these areas[39-41]. Our study builds upon this understanding by revealing that more urbanised environments, characterised by greater land-use diversity and a greater variety of amenities, may offer increased opportunities for cross-group interactions in activity sites[42,43]. These environments foster social integration by enabling encounters among individuals from diverse socioeconomic backgrounds beyond residential areas. Notably, the reduction in experienced segregation relative to residential segregation is most pronounced among neighbourhoods with high residential segregation, and those characterised by low-income and majority-POC. This suggests that access to diverse activity sites beyond immediate neighbourhoods is vital for socially disadvantaged neighbourhoods with high residential segregation, as it enhances upward economic mobility and reduces experienced segregation[13,44,45].

Second, our study demonstrates that experienced segregation is linked to travel distance from a home neighbourhood, based on intra-neighbourhood analyses. Travelling to activity sites over longer distances from one's home neighbourhood is more likely to reduce experienced segregation and help mitigate residential segregation than travelling to those over shorter distances. This effect is especially pronounced among neighbourhoods with high levels of residential

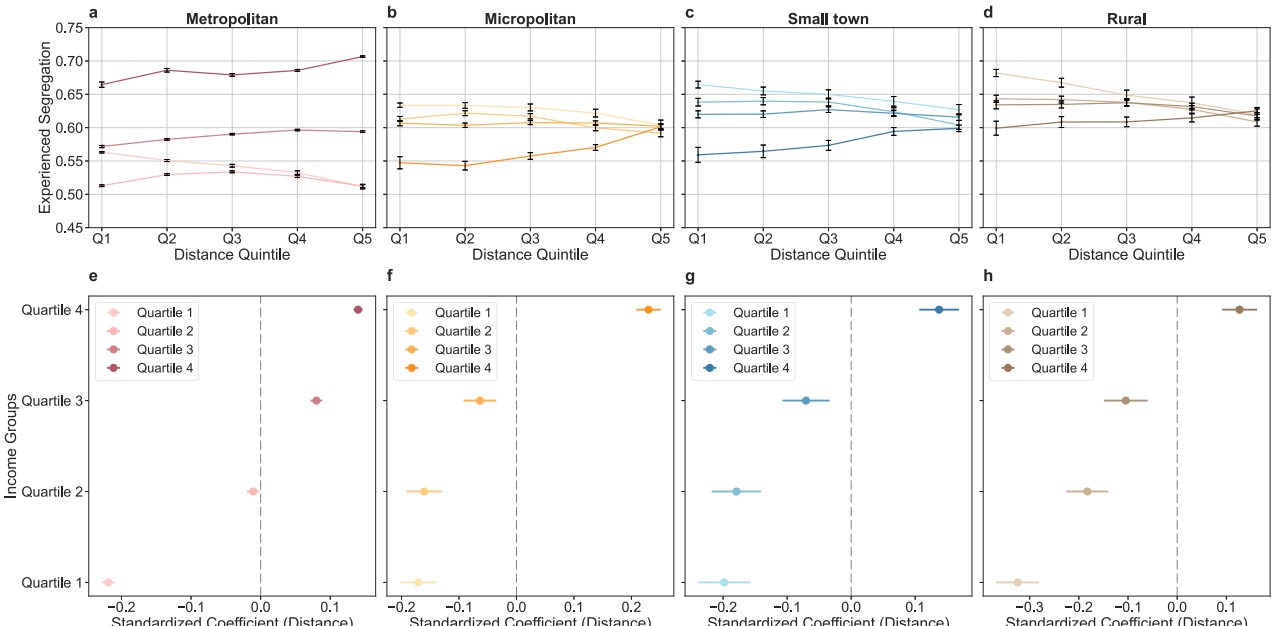

**Fig. 6 | Associations between average travel distance and experienced segregation across neighbourhoods. a–d** Experienced segregation by average travel distance across neighbourhoods in different income quartiles within metropolitan areas, micropolitan areas, small towns, and rural areas. In **a–d**, Q1, Q2, Q3, Q4, and Q5 denote travel distance quintiles 1–5, respectively. For metropolitan areas, *n* = 41,977, 41,913, 41,946, and 41,944 neighbourhoods in income quartiles 1–4. For micropolitan areas, *n* = 5146, 5144, 5145, and 5145 neighbourhoods in income quartiles 1–4. For small towns, *n* = 2713, 2711, 2711, and 2712 neighbourhoods in income quartiles 1–4. For rural areas, *n* = 2171, 2177, 2166, and 2169 neighbourhoods in income quartiles 1–4. Points show the mean experienced segregation for each income quartile across travel distance quintiles within each urbanicity level, with error bars indicating 95% confidence intervals. **e–h** Coefficients of travel distance for each income quartile in metropolitan areas (n = 167,780 neighbourhoods), micropolitan areas (n = 20,580 neighbourhoods), small towns (n = 10,847 neighbourhoods), and rural areas (n = 8683 neighbourhoods) from multiple linear regression models. The error bars indicate 95% confidence intervals.

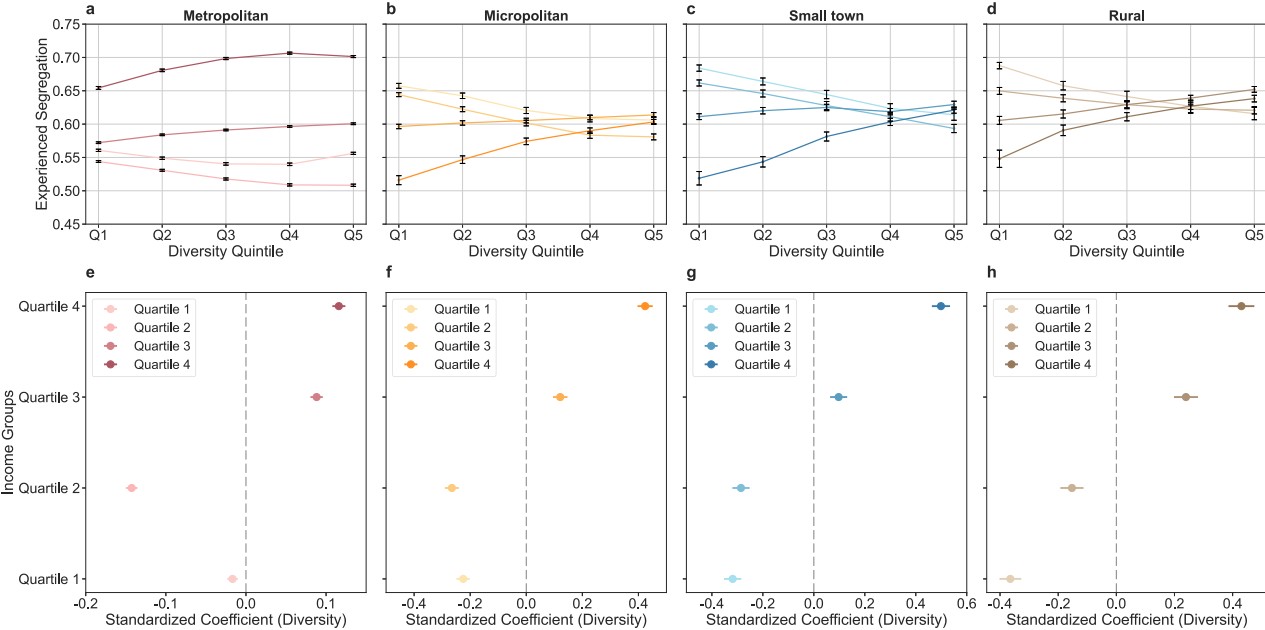

**Fig. 7 | Associations between travel diversity and experienced segregation across neighbourhoods. a–d** Experienced segregation by travel diversity across neighbourhoods in different income quartiles within metropolitan areas, micropolitan areas, small towns, and rural areas. In **a–d**, Q1, Q2, Q3, Q4, and Q5 denote travel diversity quintiles 1–5, respectively. For metropolitan areas, *n* = 41,977, 41,913, 41,946, and 41,944 neighbourhoods in income quartiles 1–4. For micropolitan areas, *n* = 5146, 5144, 5145, and 5145 neighbourhoods in income quartiles 1–4. For small towns, *n* = 2713, 2711, 2711, and 2712 neighbourhoods in income quartiles 1–4. For rural areas, *n* = 2171, 2177, 2166, and 2169 neighbourhoods in income quartiles 1–4. Points show the mean experienced segregation for each income quartile across travel diversity quintiles within each urbanicity level, with error bars indicating 95% confidence intervals. **e–h** Coefficients of travel diversity for each income quartile in metropolitan areas (*n* = 167,780 neighbourhoods), micropolitan areas (*n* = 20,580 neighbourhoods), small towns (*n* = 10,847 neighbourhoods), and rural areas (*n* = 8683 neighbourhoods) from multiple linear regression models. The error bars indicate 95% confidence intervals.

segregation, suggesting that long-distance travel may offer greater benefits of reduced income segregation for these neighbourhoods. These findings challenge the localised living models, which may work well in socially mixed neighbourhoods, where cross-class interactions are feasible even when their residents' activities are largely confined to a limited geographic area. However, in neighbourhoods characterised by high residential segregation, localised living within a socially homogeneous environment risks reinforcing segregation by restricting residents' exposure to diverse social contexts. Moreover, our findings also suggest that the benefits of long-distance trips on reduced experienced segregation are more pronounced among socially disadvantaged neighbourhoods (i.e., those with low income and a majority population of POC), especially those in less urbanised areas. This may be because residents of socially disadvantaged neighbourhoods, particularly those in less urbanised areas, often face transportation constraints and travel less frequently over long distances[46–48]. As a result, long-distance trips provide rare opportunities for exposure to more affluent and socially distinct populations, thereby enhancing the integration potential[44,49].

Third, our findings reveal how the overall experienced segregation relates to the average travel distance and travel diversity across different neighbourhoods, based on inter-neighbourhood analyses. A consistent pattern emerges, particularly in low-income neighbourhoods: those with longer average travel distance and more diverse destinations tend to experience lower levels of segregation. For low-income neighbourhoods, longer average travel distance and greater travel diversity increase the likelihood that residents to break localised living patterns and encounter individuals from more affluent backgrounds. These relationships are more pronounced in less urbanised settings, where disparities in both neighbourhood travel behaviour and experienced segregation across neighbourhoods are greater. Moreover, the negative relationships between average travel distance and experienced segregation are more pronounced in low-income neighbourhoods with high residential segregation and a majority population of POC in cities, whereas travel diversity exhibits a stronger relationship with reduced experienced segregation across low-income neighbourhoods with limited access to public transit. Conversely, residents of affluent neighbourhoods typically benefit from a wide array of local amenities and flexible travel options, allowing them to travel further without significant changes to their routines. Moreover, residents from affluent neighbourhoods often demonstrate self-segregation[39], as they are more likely to visit places mostly used by affluent individuals, e.g., golf courses, resorts, and high-end shopping malls, regardless of travel distance. As a result, people from high-income neighbourhoods, despite greater travel distances and more diverse travel patterns, may experience higher levels of income segregation due to self-selection exposure. Thus, the relationships between neighbourhood travel behaviour and experienced segregation in these neighbourhoods are weaker or even reversed.

Our findings highlight the need to consider the potential trade-off between localised living models and the risk of heightened income segregation, particularly in low-income neighbourhoods[15]. While localised living models have proven effective in compact global cities, the United States is largely characterised by urban sprawl and auto-mobile dependence, where public transit is limited and residential segregation is often entrenched[50,51]. In such contexts, localised living may inadvertently restrict residents' mobility beyond their immediate neighbourhoods, confining them to already segregated environments and reducing opportunities for cross-class social interaction[52]. This may further reinforce income segregation, particularly for socially disadvantaged neighbourhoods with high levels of residential segregation. We believe that long-distance and diverse travel to a broader range of activity sites is vital for residents of socially disadvantaged neighbourhoods with high levels of residential segregation. Such extended mobility enables these residents to break out of socially

homogeneous residential settings, engage with more diverse populations, and reduce experienced segregation. An effective strategy is to provide affordable and well-connected public transit services in less affluent neighbourhoods, which may increase mobility for residents[36,44], thereby facilitating access to diverse activity sites across a larger area. In addition, policymakers and urban planners should increase investment in inclusive and high-quality public spaces, e.g., parks, recreational plazas, and sports playgrounds, which usually have no financial barriers and accommodate people of all income groups[45]. Such initiatives can play a pivotal role in fostering social integration.

Our study is subject to several limitations. First, as mobility records are aggregated at the CBG level, we conducted analyses at the neighbourhood level rather than the individual level. However, CBGs are the smallest spatial unit with available both mobility data and socioeconomic data, providing sufficient spatial resolution and a large representative sample for neighbourhood-scale analyses. Second, given the temporal resolution of the dataset, our analyses are limited to estimating the likelihood of co-location, namely the probability that individuals from different neighbourhoods are present at the same activity site, on a weekly basis, which serves as a proxy for social exposure. However, individuals from a neighbourhood may visit the same place at different times or on different days of the week. While our approach does not capture more precise temporal overlap, weekly mobility records provide a reliable approximation of individuals' general travel patterns. Accordingly, the experienced exposure measured in this study is likely to reflect real-world exposure with reliable accuracy. Finally, it should be noted that different types of activity sites vary in their potential to facilitate social interaction and thus may contribute unequally to experienced segregation[11,53]. For instance, individuals are likely to engage in markedly different levels of interaction in interaction-restricted environments (e.g., theatres and cinemas) compared to interaction-intensive settings (e.g., schools and parks), even with equal visiting duration. Although this distinction is critical, it remains challenging to quantify due to the absence of interaction-specific information in the mobility data or previous research. We conducted a sensitivity analysis by recalculating experienced segregation using visits to eight selected types of activity sites identified in prior research as particularly conducive to social interaction. The results are highly consistent with those calculated using the full set of activity spaces, suggesting that the omission of interaction intensity across space types has minimal impact on the overall measurement of experienced segregation. Future studies are encouraged to integrate mobility data with survey-based measures of social interaction intensity across different activity spaces to improve the accuracy of experienced segregation estimates.

## Methods
### Datasets and processing
**Mobility data.** Mobility data for this study were drawn from Safe-Graph's Weekly Patterns dataset. Our sample consisted of mobility data from the contiguous United States for each week of 2019[54]. This dataset provides origin-destination (OD) travel data aggregated from approximately 10% of the anonymised mobile devices used in the United States. Each record includes location information (latitude and longitude) on destination points of interest (POIs), visitor counts, and the census block group (CBG) identifiers of the visitors' home locations. CBGs are statistical subdivisions of census tracts defined by the U.S. Census Bureau, each comprising clusters of blocks within the same tract and typically encompassing populations ranging from 600 to 3000 individuals. Home locations are determined by SafeGraph using nighttime data (6 pm to 7 am) over a 6-week period, with each device's location assigned at the geodash-7 level and mapped to CBGs. Hence, in this study, home CBG identifiers in the mobility records were used to represent the residential neighbourhoods of visitors within each OD flow. Notably, this dataset may underrepresent individuals

who infrequently use smartphones or choose not to share location data. However, research has demonstrated a strong correlation between device counts and populations across spatial scales[55], suggesting that SafeGraph data provide relatively comprehensive coverage across geographic regions and population groups—including gender, age, race, and income.

Destination POIs were used to represent the activity sites visited by individuals. Each POI was classified using six-digit North American Industry Classification System (NAICS) codes from SafeGraph's Core Places dataset. Our analyses focused on 16 representative types of activity sites, which together account for over 70% of all locations in the dataset and reflect a broad range of resident needs: basic living (markets, restaurants, life services, and hotels); health and well-being (healthcare facilities, hospitals, and personal care); cultural and spiritual engagement (cultural and religious facilities); education and social development (schools and social assistance); recreation and leisure (entertainment venues, sports facilities, and parks); and consumer and retail activities (shopping stores and groceries). The selected types of activity sites and their corresponding NAICS codes are provided in Supplementary Table 1. We then linked the SafeGraph Places dataset to the Weekly Patterns dataset using the unique SafeGraph Place IDs to identify the OD travel pairs for POIs in each type. Overall, our sample included 1,226,450,275 OD pairs across 3,227,842 POIs. The number of OD pairs for each type of activity site and the corresponding total number of POIs are presented in Supplementary Table 2.

**Definition of urbanicity.** The definition of urbanicity for each CBG was based on the 2010 Rural–Urban Commuting Area (RUCA) Codes produced by the Economic Research Service (ERS) of the United States Department of Agriculture (USDA)[56]. These codes classify urban cores and adjacent areas that are economically integrated with those cores based on measures of population density, urbanisation, and daily commuting at the census tract level. The RUCA codes are used to define 10 primary categories of urbanicity based on the size and direction of primary commuting flows. The detailed descriptions of the primary codes can be found in Supplementary Table 3. To streamline our calculations, we reclassified the census tracts into four urbanicity groups based on the primary flow codes. Metropolitan tracts included those in which the primary mobility flow was within an urbanised area (UA), 30% or more of the mobility flow was to a UA, or 10% to 30% of the mobility flow was to a UA. Micropolitan tracts encompassed those with a primary mobility flow within a large urban cluster (UC) of 10,000 to 49,999 residents, with 30% or more of the mobility flow to a large UC, or with 10% to 30% of the mobility flow to a large UC. Small town tracts were those with a primary mobility flow within a small UC of 2500 to 9999 residents, with 30% or more of the mobility flow to a small UC, or with 10% to 30% of the mobility flow to a small UC. Rural tracts were defined as those with a primary mobility flow to a tract outside a UA or UC. In this study, we assigned each CBG the same urbanicity level as the census tract in which it is located.

**Sociodemographic data.** The empirical analyses in this study were primarily conducted at the neighbourhood level, represented by CBG. We extracted sociodemographic data of CBGs from the American Community Survey (ACS) 5-Year Estimates for 2015–2019. To categorise income groups, we used the median household income data for each CBG. For the empirical analyses of differences across urbanicity levels, the income quartile of a neighbourhood was calculated based on all of the neighbourhoods in each urbanicity level. Moreover, racial composition data were used to classify neighbourhoods by racial majority: those with more than 50% White residents were defined as majority-White neighbourhoods, while those with more than 50% POC (people of colour) residents were defined as majority-POC neighbourhoods. For the statistical analyses, we complied additional sociodemographic variables, including population size, land area

(used to calculate population density), median age, the proportion of individuals with a high school diploma, the proportion of White people, the proportion of females, and the proportion of tenures with access to at least one vehicle (used as a proxy for vehicle availability). Notably, a small number of CBGs were excluded from the analyses due to missing relevant data. In total, our analyses included 207,890 CBGs, of which 167,780 were classified as metropolitan areas, 20,580 as micropolitan areas, 10,847 as small towns, and 8683 as rural areas.

**Measuring experienced income segregation**
As the mobility data are aggregated at the CBG level, income segregation was measured at the CBG level as well, which we refer to as the neighbourhood level throughout our analyses. Conventional methods for measuring segregation using categorical variables, such as racial segregation, are less applicable to income segregation, as categorising income can lead to a loss of information. In addition, individuals across the income spectrum may perceive their social proximity to others differently. For example, those in the lowest income group may feel closer to middle-income individuals than to those at the top of the distribution, while middle-income individuals may perceive themselves as most socially distant from both extremes. To account for such nuances, we adopted the concept of 'income distance' developed by Xu et al.[34], which quantifies the perceived proximity between individuals of different income levels. The concept was originally developed to quantify perceived proximity between individuals with differing income levels. As income data are only available at the CBG level and the mobility data used in this study are aggregated at the same spatial scale, we extended this approach by calculating income distance between CBGs. To do so, we first ranked all CBGs by their median household income, from lowest to highest, assigning each a rank to represent its relative income position.

$$(R_n)_{n=1}^N = (1, 2, 3, \ldots, N) \tag{1}$$

Where $R$ is the income rank of each CBG and $N$ is the number of CBGs. We used median household income rather than average income for ranking, as the median is more robust to extreme values and better reflects the typical income level within a neighbourhood. The naïve income distance between two CBGs can be defined as the absolute difference in their income ranks. Building on this concept, we constructed an income dissimilarity matrix to quantify the relative income dissimilarity between each pair of CBGs in our sample. Specifically, for any two CBGs, $CBG_i$ and $CBG_j$, we assumed the existence of a set of $K$ 'mediator CBGs', denoted as $CBG_k$, which are closer (i.e., with smaller income distance) to $CBG_i$ than $CBG_i$ is to $CBG_j$:

$$K = \{k \,|\, |R_k - R_i| < |R_i - R_j|\} \tag{2}$$

The income dissimilarity from $CBG_i$ to $CBG_j$ is then defined as:

$$D_{i \to j} = \begin{cases} \frac{|K| + 0.5}{N-1}, & \text{if there exists another } l (l \neq j) \text{ such that } |R_l - R_i| = |R_i - R_j| \\ \frac{|K|}{N-1}, & \text{otherwise} \end{cases} \tag{3}$$

Where $|K|$ is the cardinality of $K$. This income dissimilarity is thus referred to as the proportion of CBGs that are closer to $CBG_i$ than $CBG_i$ is to $CBG_j$, relative to the total number of compared CBGs. It is worth noting that $D_{i \to j}$ might not be equal to $D_{j \to i}$, reflecting the differing perceptions of individuals with varying income levels. $D_{i \to j}$ ranges from 0 to 1, with a higher value indicating a higher level of income dissimilarity. We assumed that individuals exposed to people from neighbourhoods with more dissimilar income levels than their own would experience lower levels of segregation compared to those exposed to individuals from income-similar neighbourhoods.

Following the approach of Xu et al.[34], the experienced income dissimilarity $E_{k_i, n, w}$ at an activity site (represented by POI) $n$ of any individual from home $CBG_i$ was defined as the income dissimilarity between that individual and the visitors from other CBGs over the course of a week, weighted by the number of visitors from other CBGs to account for the probability of exposure to visitors from different CBGs:

$$E_{(k_i, n, w)} = \frac{\sum_j V_{j, n, w} \times D_{i \to j}}{\sum_j V_{j, n, w}} \qquad (4)$$

where $V_{j, n, w}$ is the number of visitors from $CBG_j$ at POI $n$ in week $w$.

By extending the metric from the individual to the neighbourhood level, the overall experienced income dissimilarity for $CBG_i$ in week $w$ was then defined as the weighted average of income dissimilarity experienced by its residents across all activity sites, with weights determined by the number of visitors from $CBG_i$ to each space:

$$E_{i, w} = \frac{\sum_n V_{i, n, w} \times E_{(k_i, n, w)}}{\sum_n V_{i, n, w}} \qquad (5)$$

where $V_{i, n, w}$ is the number of visitors from $CBG_i$ to POI $n$ in week $w$. The experienced segregation for $CBG_i$ during the entire year was then calculated as:

$$S_i = 1 - \frac{\sum_w V_{i, w} \times E_{i, w}}{\sum_w V_{i, w}} \qquad (6)$$

where $V_{i, w}$ is the number of visitors from $CBG_i$ in week $w$. Experienced segregation ranges from 0 to 1, with the value closer to 1 indicating that individuals from a neighbourhood are predominantly exposed to others from neighbourhoods with similar income levels, and values closer to 0 indicating greater exposure to individuals from income-dissimilar neighbourhoods. We also calculated, for each neighbourhood, the experienced segregation associated with visits to each type of activity site, based on mobility records linked to POIs within that category.

Notably, the SafeGraph Weekly Pattern dataset used in this study does not allow for the precise measurement of interactions among individuals, as visitor counts are aggregated on a weekly basis. Consequently, we cannot identify different individuals who simultaneously visited the same locations or determine the duration of their visits. However, weeks represent the finest temporal resolution available in the SafeGraph dataset, and the visitor-weighted method effectively captures the likelihood that individuals from different neighbourhoods encounter one another by sharing the same activity site within a given week. This approach has been established as a reliable proxy for measuring social exposure and has been widely adopted in existing studies[30,44,57]. To assess the robustness of our approach, we re-performed the calculation by incorporating the median dwell time for each activity site, thereby accounting for variation in the intensity of exposure across different activity sites. Specifically, we recalculated the overall experienced income dissimilarity for $CBG_i$ in week $w$ by weighting experienced income dissimilarity at each activity site by the product of the number of visitors and the median dwell time at POI $n$. The experienced segregation for $CBG_i$ over the entire year was then recalculated as:

$$S_i' = 1 - \frac{\sum_w \sum_n (V_{i, n, w} \cdot T_{n, w}) \times E_{(k_i, n, w)}}{\sum_w \sum_n (V_{i, n, w} \cdot T_{n, w})} \qquad (7)$$

where $T_{n, w}$ is the median dwell time for POI $n$ in week $w$. $S_i'$ is highly correlated with $S_i$ (Spearman correlation = 0.99, $p < 0.001$), indicating strong consistency and robustness of our results (Supplementary Fig. 27). The high correlation between two indices is consistent across all type of activity sites, with Spearman correlation ranging from 0.84

in life service settings ($p < 0.001$) to 1.00 in hospitals ($p < 0.001$) (Supplementary Fig. 28). Notably, not all activity sites contribute equally to social interaction and, by extension, to experienced segregation. For instance, interactions may be limited in some settings, e.g., theatres and cinemas, but more frequent in socially intensive environments such as schools. Although our primary approach does not differentiate activity sites by their propensity to facilitate interaction, we conducted another sensitivity analysis by recalculating experienced segregation using a subset of eight activity site types identified as more conducive to social interaction, including groceries, markets, parks, religious venues, restaurants, schools, sport facilities, and shopping stores[58,59]. The resulting measure $S_{i, selected}$ shows an almost perfect correlation with the original index $S_i$ (Spearman correlation = 1.00, $p < 0.001$), suggesting that the influence of excluding this differentiation on the overall estimates is negligible (Supplementary Fig. 29). This conclusion is further supported by the strong correlation between $S_{i, selected}'$, which additionally accounts for median dwell time, and the original $S_i$ (Spearman correlation = 0.99, $p < 0.001$) (Supplementary Fig. 30).

## Measuring residential income segregation
We further measured residential segregation using the same income dissimilarity matrix to examine its relationship with experienced segregation. We defined overall residential income dissimilarity for each neighbourhood as the population-weighted average income dissimilarity between the focal neighbourhood and its adjacent neighbours. The adjacent neighbour was determined using the queen contiguity method, which considers two neighbourhoods adjacent if their boundaries share either a common edge or vertex. The residential segregation of $CBG_i$ was then calculated as:

$$S_{i, residential}' = 1 - \frac{\sum_j P_j \times D_{i \to j}}{\sum_j P_j} \qquad (8)$$

Where $P_j$ is the population of the adjacent neighbourhood $CBG_j$ relative to $CBG_i$.

## Measuring travel distance
In this study, the travel distance of a given CBG (defined as a neighbourhood) was defined as the visitor-weighted average distance of all OD pairs associated with that CBG. To calculate this, we extracted the latitudes and longitudes of the centroids of all of the CBGs using boundary shapefiles from the Topologically Integrated Geographic Encoding and Referencing System geodatabases. For each OD pair in the mobility dataset, we assessed the Euclidean distance between the home CBG and the destination POI. The weighted average travel distance was then calculated for each CBG, with weights determined by the number of visitors from the given CBG to each destination POI throughout the year[36]:

$$\text{Distance}_i = \frac{\sum_w \sum_n V_{i, n, w} \times \text{Distance}_{i \to n}}{\sum_w \sum_n V_{i, n, w}} \qquad (9)$$

where $V_{i, n, w}$ is the number of visitors from $CBG_i$ to POI $n$ in week $w$, and $\text{Distance}_{i \to n}$ is the Euclidean distance between $CBG_i$ and POI $n$.

## Decomposing experienced segregation by travel distance
We adopted two complementary approaches to decompose experienced segregation and assess disparities in experienced segregation across trips with different travel distances. First, we decomposed experienced segregation based on the trips inside and outside the home neighbourhood, recognising that trips outside the home neighbourhood typically cover greater distances than those inside the home neighbourhood. For each neighbourhood, we used trips to activity sites located within the home neighbourhood to assess inside-

neighbourhood segregation, and trips to activity sites situated outside the home neighbourhood to measure outside-neighbourhood segregation. Second, we decomposed experienced segregation based on the travel distance of each trip. We categorised trips by travel distance, assigning distance quartiles separately for trips related to neighbourhoods within each urbanicity level. For each neighbourhood, experienced segregation was calculated for trips within each distance quartile.

## Measuring travel diversity

To measure travel diversity, we first calculated the proportion of visitors from each neighbourhood to each type of activity site. We then used the entropy method to quantify the extent to which visitors equally visit different types of activity sites:

$$\text{Diversity}_i = \frac{\sum_i r_i \ln(r_i)}{-\ln(16)} \tag{10}$$

where $r_i$ is the ratio of visitors to the $i^{th}$ ($i = 1, 2, 3, \ldots, 16$) category of activity sites. Diversity$_i$ ranges from 0 to 1, where 0 indicates that all visits from a neighbourhood are concentrated in a single type of activity site (low diversity), while a value of 1 reflects an equal distribution of visits across all activity site types (high diversity).

## Inter-neighbourhood analyses

We finally conducted inter-neighbourhood analyses to explore the relationship between neighbourhood travel behaviour and experienced segregation across neighbourhoods. For the overall comparison, we conducted multivariate linear regressions to examine the relationships between travel distance, travel diversity, and experienced segregation across neighbourhoods with different income levels within each urbanicity, respectively. Given the notable differences in income levels, travel distance, travel diversity, and experienced segregation among neighbourhoods with different urbanicity levels, we regressed experienced segregation on travel distance and travel diversity separately for the neighbourhoods of each urbanicity level (i.e., metropolitan, micropolitan, small town, and rural). Income quartile was included as a categorical variable in the model, along with an interaction term for "Income Quartile × Travel Distance" and "Income Quartile × Travel Diversity" to assess whether the associations between travel distance, travel diversity, and experienced segregation differed among neighbourhoods at varying income levels. The sociodemographic variables mentioned in the socioeconomic data section were included in the models as covariates to account for the potential impacts of age, gender, race, education, population size, and vehicle availability. The independent and dependent variables included in the models were standardised before the regression analyses. To test for multicollinearity among the selected variables, we assessed the variance inflation factor (VIF) for all variables in the full models. The results indicated that the VIFs for all variables were less than 4, suggesting that there were no multicollinearity issues in the model.

Moreover, we classified neighbourhoods in income quartiles 1 and 2 as low-income and conducted subsample analyses restricted to this group to explore whether the associations between experienced segregation and neighbourhood travel behaviour vary across low-income neighbourhoods with different characteristics. Three sets of subsample analyses were performed based on key neighbourhood characteristics: (1) high versus low levels of residential segregation, (2) majority-POC versus majority-White neighbourhoods, and (3) neighbourhoods with versus without access to public transit within a 15-min walking distance. Neighbourhoods were classified as high or low in residential segregation based on whether their values were above or below the median residential segregation of sample within each urbanicity level, respectively. Racial majority was determined by

population composition, with neighbourhoods comprising more than 50% White residents designated as majority-White neighbourhoods, and the rest as majority-POC neighbourhoods. Access to public transit was identified using the Smart Location Database, categorising neighbourhoods based on the presence or absence of public transit stops within a 15-min walking distance.

## Reporting summary

Further information on research design is available in the Nature Portfolio Reporting Summary linked to this article.

## Data availability

The mobility data was obtained from SafeGraph, which is commercially available and can be requested for research use (https://www.safegraph.com/pricing). Other data used in this study are open access. The NAICS codes for POIs classification were sourced from the official website for census bureau of the United States (https://www.census.gov/naics/). The socioeconomic data was derived from the American Community Survey (ACS) 5-Year Estimates for 2015–2019 (https://data.census.gov/). The RUCA codes for defining urbanicity levels were sourced from the USDA. Shapefiles for mapping were obtained from the Topologically Integrated Geographic Encoding and Referencing System (TIGER) geodatabases (https://www.census.gov/geographies/mapping-files/time-series/geo/tiger-geodatabase-file.html). Data of access to public transit was derived from Smart Location Database provided by Environmental Protection Agency (EPA) of the United States (https://www.epa.gov/smartgrowth/smart-location-mapping). Source Data are provided in the following repository: https://figshare.com/projects/Travel_Behaviour_and_Income_Segregation/266242 (ref. 60).

## Code availability

All codes used to produce main and supplementary results in this study are available in the following repository: https://figshare.com/projects/Travel_Behaviour_and_Income_Segregation/266242 (ref. 60).

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

## Acknowledgements
The authors thank Dr. Yang Xu from The Hong Kong Polytechnic University for his valuable guidance on the methodology used in this study.

## Author contributions
Y.Z. designed the study, finalised the analysis, interpreted the findings and wrote the paper. Y.L. designed the study, collected data, interpreted the findings, commented on and revised drafts of the paper.

## Competing interests
The authors declare no competing interests.
