## [Transparent Peer Review file · Nature Communications]

Varying relationships between experienced income segregation and travel behaviour across neighbourhood social and urban contexts

Corresponding Author: Professor Yi Lu

Version 0:

Reviewer comments:

Reviewer #1

(Remarks to the Author)

This paper uses mobile phone-based mobility data to examine experienced income segregation at the U.S. census block group level. The study finds that block group-based experienced income segregation is relatively lower for large cities (i.e., metropolitan areas). The study also runs regressions and finds that, for residents of the least affluent communities, longer travel distances and more diverse destinations are associated with lower segregation.

I have the following comments regarding this manuscript:

1. Although in the title and many parts of the manuscript the authors refer to "socioeconomic segregation" or "SES segregation," the study only uses the relative rank of block group-level median household income as the measurement. Thus, I suggest the authors consider using "income segregation" instead of "SES segregation."
2. Line 92: Why use "median income" (which I assume refers to median household income) rather than *average* household income? I think some explanation would be helpful.
3. Regarding the development of socioeconomic distance, I find that the equation is basically the same as (or at least very similar to) reference 32 (Xu et al., 2019) – the only difference is that Xu et al. (2019) uses individuals as the analysis unit, whereas this paper uses census block groups. I suggest that the authors (1) more clearly acknowledge that the method largely follows Xu et al. (2019), rather than simply citing it in the "Methods" section, and (2) briefly discuss the differences between their method and that of Xu et al. (2019).
4. Following my previous comment, in line 422: "this study employed a mobility-based method that is able to capture the socioeconomic distance between individuals to measure socioeconomic segregation (Xu et al., 2019: ref 32)." Xu et al. (2019) did capture the SES distance between individuals (based on housing price rankings), while this study is more likely capturing income differences at the census block group (i.e., neighborhood) level. Thus, I suggest the authors adjust the wording accordingly.
5. Line 53: Since census block groups are not the smallest geographic scale for U.S. censuses, I would like to seek clarification on whether household income information is available at the census block level (from the American Community Survey).
6. Line 54: Regarding "census data," since the 2010 U.S. census no longer includes socio-economic information such as income, please replace it with "American Community Survey data."
7. The study examines differences in experienced income segregation by income groups (Figure 1, Figure 3, Figure 4, Figure 5). I wonder if it would also be informative to conduct a subsample analysis by the racial profiles of communities. For instance, simply comparing white-majority neighborhoods versus non-white-majority neighborhoods would be helpful.
8. Would the relationship between travel distance/diversity and experienced income segregation also differ based on the

level of transport infrastructure? Theoretically, the lowest income groups might have lower vehicle ownership rates and might be more reliant on public transport. Thus, examining differences by public transport availability could provide further implications for policymakers.

9. For Figure 1a, please provide the range of the experienced segregation index for each category, e.g., Low (value ~ value), etc.

10. For Figures 1b, 1c, and 1d, please provide a metropolitan-area-wide experienced segregation index for the Los Angeles, Chicago, and Houston metropolitan areas.

Reviewer #2

(Remarks to the Author)

Thank for the opportunity to review the paper „Varying relationships between travel behaviour and socioeconomic segregation across income groups and urbanicity “. This is an important paper can potentially make a significant contribution to understanding the intersection of travel behavior and socioeconomic segregation based on the study of different income groups and urban settings in the US. Using a vast dataset of 1.2 billion mobility records, the study estimates individuals' experienced socioeconomic segregation and examines how travel distance and diversity impact segregation levels among various income groups and urbanicity categories. The study's key finding is that longer travel distances and more diverse destinations reduce segregation for residents of low-income communities, particularly in less urbanized areas, while the effect is less evident or even opposite for higher-income groups.

1. The text is too technical, using terms like "CBGs" instead of "neighborhood" and "POI" instead of "activity site." It would be more effective to use conceptually relevant terms and clarify how they are technically measured (e.g., by CBGs or POIs) in this specific study.

2. Both conceptually and analytically, the paper does not adequately differentiate between residential segregation and non-residential segregation when moving around. This distinction is critical given the study's objective of problematizing the 15-Minute City concept and its underlying focus on promoting localized living. While the 15-Minute City concept may not be problematic in residentially mixed cities, it could become highly problematic in cities with pronounced residential segregation. Since neighborhood income context data is available, the study could examine the relationship between residential and non-residential segregation across different urban contexts and travel distances.

3. The study does not incorporate well the key concepts "activity space", and does not relate to more recent conceptual advancements such as "cocooning," "multicontextual segregation," "domains of segregation," or "vicious circles of segregation." POI is a technical term, but "activity site" is the underlying conceptual construct running through all these frameworks, but is missing in this study. Like-wise, „co-presence“ and „co-location“ should be conceptualised.

4. Not all activity sites and related activities have the same significance in terms of experienced segregation. For example, schools and cinemas have different dynamics of social interaction. It would be beneficial to relate the study to activity-based research to capture these distinctions more effectively.

5. The study lacks clarity on the type of contextual information available about activity sites, such as schools. How exactly does the study capture the residential context, the context of other activity sites, the spatial resolution, and the duration of stay in these sites? This information is necessary to understand the study's contribution comprehensively.

6. Although methodological details are provided in a separate section, a nutshell information should be available earlier in the manuscript. For instance, the first paragraph in the "Results" section states that "the overall median experienced segregation across all CBGs was 0.594." Without any context, this statement does not carry meaning. It is necessary to clarify, in nutshell: (a) what CBGs represent, (b) the groups being studied, (c) what is experienced segregation, and (d) how what is the measure used, including its meaning and range.

7. The finding that experienced segregation decreases with increasing metropolitan size is unexpected and contradicts much of the existing literature on residential segregation by city size. This result requires more extensive discussion, linking to previous studies and exploring possible explanations. May it be due to mobility patterns, increased access to diverse activity sites, or other factors?

8. The results on trip length and segregation are important. However, it would be valuable to further explore whether longer trips specifically reduce experienced segregation in highly segregated cities, potentially revealing nuanced dynamics of residential and non-residential segregation.

9. The Discussion section should link back to the 15-Minute City concept and localized living in different urban contexts. Also, the discussion of the specific context of the US with large sprawled city-regions and automobile-based travel should be addressed to increase the international impact. These extensions could be made by significantly cutting back and streamlining existing discussions.

Version 1:

Reviewer comments:

Reviewer #1

(Remarks to the Author)

I would like to thank the authors for addressing my previous comments. I have the following remaining – hopefully minor – comments, which I hope the authors could also address:

1. Regarding my previous comment #7 (subsample analysis by racial profile of communities): as shown in Supplementary Figures 3-4 in this revision, the levels of experienced segregation between white-majority and non-white-majority communities differ between metropolitan areas and the other three settlement types (i.e., micropolitan, small town, and rural). What could explain this pattern? Could it be due to differences in trip distance, or something else? Some discussions on this would enhance the paper.

2. For my previous comment #9 (range of experienced segregation index in the map): in the revised Figure 1a, the legend is currently written as follows—low (≤ 0.529), moderate low (≤ 0.570), etc. I believe this should be corrected for clarity and accuracy. For example: low (≤ 0.529), moderate low (0.530~0.570), and so on. Please update this accordingly.

3. I am not sure whether the term “segregation reduction”, as used in Supplementary Figure 6 and elsewhere in the manuscript, is appropriate. The term “reduction” typically implies temporal change or causal inference, which does not appear to be the case in this context. Perhaps a term like “residential-experience segregation disparity” or something similar would be more suitable.

Reviewer #2

(Remarks to the Author)

Thank you very much, the response letter is very well organized, and I am happy with the thorough and thoughtful revisions.

Response letter

We sincerely thank the editor and both reviewers for their thoughtful and constructive feedback, which significantly helped us improve the quality and depth of our manuscript. In response to the editorial requirements and reviewer suggestions, we conducted two additional sensitivity and uncertainty analyses, which further demonstrate the robustness of our study. Moreover, we incorporated three additional analyses, including residential segregation, racial disparities, and public transit analysis, which enhance the depth of study. We also restructured the discussion accordingly. Below, we provide a detailed, point-by-point response to each reviewer's comments. Reviewer comments are shown in black, our responses in blue, and the corresponding revisions in the manuscript in red.

Reviewer #1 (Remarks to the Author):

This paper uses mobile phone-based mobility data to examine experienced income segregation at the U.S. census block group level. The study finds that block group-based experienced income segregation is relatively lower for large cities (i.e., metropolitan areas). The study also runs regressions and finds that, for residents of the least affluent communities, longer travel distances and more diverse destinations are associated with lower segregation.

Thank you very much for your detailed assessment and the above summary.

I have the following comments regarding this manuscript:

1. Although in the title and many parts of the manuscript the authors refer to “socioeconomic segregation” or “SES segregation,” the study only uses the relative rank of block group-level median household income as the measurement. Thus, I suggest the authors consider using “income segregation” instead of “SES segregation.”

Response: Thank you for the suggestion. We agree that ‘income segregation’ is more accurate, and we have replaced “SES segregation” with “income segregation” throughout the manuscript.

2. Line 92: Why use “median income” (which I assume refers to median household income) rather than *average* household income? I think some explanation would be helpful.

Response: Thank you for this comment. We have clarified our methodological description to explicitly state that experienced segregation is measured using median household income, as now indicated in the opening paragraph of the Results section (Line 108-110):

“To quantify experienced segregation for residents of each neighbourhood, we first constructed an income dissimilarity matrix using the ranked median household income of each neighbourhood, ...”

We used median household income rather than the average value because the former is less sensitive to extreme values or outliers in the income distribution. Moreover, median household income provides a representative measure of neighbourhoods' general income levels, which is

thus more appropriate for quantifying experienced segregation at the neighbourhood level. We have added the explanation in the Methods section (Line 600-602):

“We used median household income rather than average income for ranking, as the median is more robust to extreme values and better reflects the typical income level within a neighbourhood.”

3. Regarding the development of socioeconomic distance, I find that the equation is basically the same as (or at least very similar to) reference 32 (Xu et al., 2019) – the only difference is that Xu et al. (2019) uses individuals as the analysis unit, whereas this paper uses census block groups. I suggest that the authors (1) more clearly acknowledge that the method largely follows Xu et al. (2019), rather than simply citing it in the “Methods” section, and (2) briefly discuss the differences between their method and that of Xu et al. (2019).

Response: Thank you for this valuable suggestion. In response, we have clearly acknowledged that we adopted the approach of Xu et al. to measure experienced segregation. We have also clarified how this method was extended to the CBG level, as now detailed in the Methods section (Line 591-597):

“To account for such nuances, we adopted the concept of ‘income distance’ developed by Xu et al. ³⁴, which quantifies the perceived proximity between individuals of different income levels. The concept was originally developed to quantify perceived proximity between individuals with differing income levels. As income data are only available at the CBG level and the mobility data used in this study are aggregated at the same spatial scale, we extended this approach by calculating income distance between CBGs.”

4. Following my previous comment, in line 422: "this study employed a mobility-based method that is able to capture the socioeconomic distance between individuals to measure socioeconomic segregation (Xu et al., 2019: ref 32)." Xu et al. (2019) did capture the SES distance between individuals (based on housing price rankings), while this study is more likely capturing income differences at the census block group (i.e., neighborhood) level. Thus, I suggest the authors adjust the wording accordingly.

Response: Thank you for this suggestion. We acknowledge that income distance was measured at the CBG level, as income data are only available at this spatial scale and the mobility data are likewise aggregated at the CBG level (see the revision in the response above).

5. Line 53: Since census block groups are not the smallest geographic scale for U.S. censuses, I would like to seek clarification on whether household income information is available at the census block level (from the American Community Survey).

Response: Thank you for this comment. Upon reviewing the official documentation from the U.S. Census Bureau, we confirm that census blocks represent the smallest geographic units for which decennial census data are collected and tabulated. However, household income data are not available at the block level. Instead, income information is reported at the census block group (CBG) level, the smallest unit for which both income and mobility data are available. We

have clarified this point in the Results section accordingly (Line 105-109).

“Even though CBGs are the next level above census blocks, income data from ACS are not available at the block level, CBGs are thus the smallest geographic units for which mobility and income data are available. We conducted our analyses at the CBG level, which we refer to as neighbourhood in this study.”

6. Line 54: Regarding "census data," since the 2010 U.S. census no longer includes socio-economic information such as income, please replace it with "American Community Survey data."

Response: Thank you for this important clarification. We have replaced all statements of “census data” with “American Community Survey data” throughout the manuscript

7. The study examines differences in experienced income segregation by income groups (Figure 1, Figure 3, Figure 4, Figure 5). I wonder if it would also be informative to conduct a subsample analysis by the racial profiles of communities. For instance, simply comparing white-majority neighborhoods versus non-white-majority neighborhoods would be helpful.

Response: Thank you for this valuable suggestion. In response, we conducted additional analyses comparing results across white-majority and non-white majority neighbourhoods. In addition to racial classification, we also classified neighbourhoods into four income-racial groups. Neighbourhoods in the 1st and 2nd income quartiles were designated as low-income group, and those in the 3rd and 4th quartiles as high-income group. These were then cross-classified by racial majority into: (1) low-income, non-white-majority; (2) low-income, white-majority; (3) high-income, non-white-majority; and (4) high-income, white-majority neighbourhoods. Results of the additional analyses are shown as follows:

1) Overall patterns of experienced segregation (Line 143-148):

“In addition to income disparities, we also examined racial disparities in experienced segregation. In metropolitan areas, neighbourhoods with non-white majorities, particularly those that are also low-income (neighbourhoods in the 1st and 2nd income quartiles), tend to exhibit relatively low levels of experienced segregation, whereas these same groups experience relatively higher income segregation in less urbanized areas (**Supplementary Fig. 3 and Supplementary Fig. 4**).”

Supplementary Fig. 3 | Distribution of overall experienced segregation between non-white and white-majority neighbourhoods across urbanicity levels

Supplementary Fig. 4 | Distribution of overall experienced segregation across income-racial groups. Neighbourhoods in the 1st and 2nd income quartiles were designated as low-income, and those in the 3rd and 4th quartiles as high-income. These were then cross-classified by racial majority into: (1) low-income, non-white-majority; (2) low-income, white-majority; (3) high-income, non-white-majority; and (4) high-income, white-majority neighbourhoods.

2) Comparison of experienced and residential segregation (Line 192-197):

“For racial disparities, non-white majority neighbourhoods, particularly those that are also low-income, present greater segregation reduction across urbanicity levels (**Supplementary Fig. 6** and **Supplementary Fig. 7**). This indicates that socially disadvantaged neighbourhood (i.e., non-white majority neighbourhoods with low income) experience greater reduced income segregation in activity sites compared to residential areas, especially in metropolitan areas.”

Supplementary Fig. 6 | Segregation reduction between non-white and white majority neighbourhoods across urbanicity levels. Reduction in experienced segregation relative to residential segregation across urbanicity levels were calculated as $(\text{residential segregation} - \text{experienced segregation}) / \text{residential segregation}$.

Supplementary Fig. 7 | Segregation reduction between across income-racial groups. Reduction in experienced segregation relative to residential segregation across urbanicity levels were calculated as $(\text{residential segregation} - \text{experienced segregation}) / \text{residential segregation}$.

3) Experienced segregation by trips of varying travel distances (Line 226-229, Line 250-252, Line 264-266, Line 288-291):

“Moreover, non-white majority neighbourhoods with low income present the greatest reduction in outside-neighbourhood trips related to inside-neighbourhood trips (Supplementary Fig. 10-Supplementary Fig. 11). Notably, this disparity becomes more pronounced as urbanicity decreases.”

Supplementary Fig. 10 | Experienced segregation for inside-neighbourhood and outside-neighbourhood trips between non-white and white majority neighbourhoods across urbanicity levels.

Supplementary Fig. 11 | Experienced segregation for inside-neighbourhood and outside-neighbourhood trips by income-racial groups across urbanicity levels.

“Similarly, greater reductions in experienced segregation with increasing travel distance are also observed among non-white majority neighbourhoods, especially those that are low-income (Supplementary Fig. 13 and Supplementary Fig. 14).”

Supplementary Fig. 13 | Patterns of experienced segregation decomposed by travel distance between non-white and white majority neighbourhoods across urbanicity levels.

Supplementary Fig. 14 | Patterns of experienced segregation decomposed by travel distance across income-racial groups and urbanicity levels.

“Moreover, non-white majority neighbourhoods, particularly those that are also low-income, experience stronger segregation reduction during longer trips than shorter ones (Supplementary Fig. 16 and Supplementary Fig. 17).”

Supplementary Fig. 16 | Reduction in experienced segregation relative to residential segregation by travel distance quartiles between non-white and white majority neighbourhoods across urbanicity levels. Noted that Only neighbourhoods where overall experienced segregation is lower than residential segregation were included in the calculation.

Supplementary Fig. 17 | Reduction in experienced segregation relative to residential segregation by travel distance quartiles across income-racial groups and urbanicity levels. Noted that Only neighbourhoods where overall experienced segregation is lower than residential segregation were included in the calculation.

“The associations are particularly pronounced for low-income neighbourhoods in less urbanized areas, especially those with a non-white majority and high residential segregation level (Supplementary Fig. 18–20)”

Supplementary Fig. 18 | Associations between mean experienced segregation and mean travel distance at the level of activity sites between non-white and white majority neighbourhoods across urbanicity levels. The texts show results of spearman correlations between experienced segregation and travel distance for activity sites ($n = 16$). Each dot represents a specific type of activity site within an income quartile, and the lines indicate the linear associations corresponding to each income quartile.

Supplementary Fig. 19 | Associations between mean experienced segregation and mean travel distance at the level of activity sites across income-racial groups and urbanicity levels. The texts show results of spearman correlations between experienced segregation and travel distance for activity sites ($n = 16$). Each dot represents a specific type of activity site within an income quartile, and the lines indicate the linear associations corresponding to each income quartile.

Supplementary Fig. 20 | Associations between mean experienced segregation and mean travel distance at the level of activity sites by income and residential segregation levels across urbanicity levels. The texts show results of spearman correlations between experienced segregation and travel distance for activity sites (n = 16). Each dot represents a specific type of activity site within an income quartile, and the lines indicate the linear associations corresponding to each income quartile.

4) Neighbourhood travel behaviour and experienced segregation (Line 365-369, Line 375-382)

“To further investigate heterogeneity in the observed associations, we conducted subsample analyses exclusively for low-income neighbourhoods (i.e., those in the 1st and 2nd income quartiles), examining whether the negative relationships between travel distance, travel diversity, and experienced segregation differ across neighbourhoods with varying levels of residential segregation, racial majorities, and access to public transit.” ...

... “Regarding racial disparities, low-income, non-white majority neighbourhoods exhibit greater reductions in experienced segregation with increasing travel distances compared to their white-majority counterparts in metropolitan areas and small towns, while this pattern is not significant among micropolitan and rural neighbourhoods (**Supplementary Fig. 23**). In contrast, low-income white-majority neighbourhoods show greater reductions in experienced segregation with increasing travel diversity across metropolitan, micropolitan, and small-town areas (**Supplementary Fig. 24**).”

Supplementary Fig. 23 | Associations between mean travel distance and experienced segregation across low-income neighbourhoods with non-white and white majorities. a-d. Experienced segregation by mean travel distance between low-income neighbourhoods with non-white and white majorities across urbanicity levels. **e-h.** The coefficients of travel distance for each group across urbanicity levels from multiple linear regression models ($n = 83, 879$ CBGs for metropolitan areas, $n = 10,279$ for micropolitan areas, $n = 5,413$ for small towns, and $n = 4,337$ for rural areas). The dark colours indicate high income quartiles, and the error bars indicate 95% confidence intervals.

Supplementary Fig. 24 | Associations between travel diversity and experienced segregation across low-income neighbourhoods with non-white and white majorities. a-d. Experienced segregation by travel diversity between low-income neighbourhoods with non-white and white majorities across urbanicity levels. **e-h.** The coefficients of travel diversity for each group across urbanicity levels from multiple linear regression models ($n = 83, 879$ CBGs for metropolitan areas, $n = 10,279$ for micropolitan areas, $n = 5,413$ for small towns, and $n = 4,337$ for rural areas). The dark colours indicate high income quartiles, and the error bars indicate 95% confidence intervals.

8. Would the relationship between travel distance/diversity and experienced income segregation also differ based on the level of transport infrastructure? Theoretically, the lowest income groups might have lower vehicle ownership rates and might be more reliant on public transport. Thus, examining differences by public transport availability could provide further implications for policymakers.

Response: Thank you for this insightful suggestion. We agree that disparities in vehicle

ownership between low-income and high-income neighbourhoods may shape patterns of experienced income segregation. To account for this, we included vehicle availability as a control variable in our statistical models, measured by the proportion of tenures with access to at least one vehicle (Line 574-578).

“For the statistical analyses, we compiled additional sociodemographic variables, including ..., and the proportion of tenures with access to at least one vehicle (used as a proxy for vehicle availability).”

In addition, we conducted a sub-sample analysis for low-income neighbourhoods to examine whether the relationship between experienced segregation and neighbourhood travel behaviour differs by access to public transit (Line 382-392).

“For public transits, while low-income neighbourhoods lacking public transit within a 15-minute walking distance show higher experienced segregation, only those in rural areas experience significantly greater segregation reduction with increasing travel distance (Supplementary Fig. 25). Across all urbanicity levels, low-income neighbourhoods without nearby public transit consistently exhibit greater reductions in experienced segregation as travel diversity increases (Supplementary Fig. 26), suggesting that neighbourhoods without accessible public transit are more sensitive to the relationship travel diversity and experienced segregation. These results suggest that access to nearby public transit plays a key role in supporting longer travel distances that mitigate experienced segregation in rural low-income neighbourhoods, and in promoting travel diversity and reducing experienced segregation in low-income neighbourhoods across urbanicity levels.”

Supplementary Fig. 25 | Associations between mean travel distance and experienced segregation across low-income neighbourhoods with and without public transit stops within a 15-minute walking distance. a-d. Experienced segregation by mean travel distance between low-income neighbourhoods with and without 15-minute public transits across urbanicity levels. **e-h.** The coefficients of travel distance for each group across urbanicity levels from multiple linear regression models ($n = 83, 879$ CBGs for metropolitan areas, $n = 10,279$ for micropolitan areas, $n = 5,413$ for small towns, and $n = 4,337$ for rural areas). The dark colours indicate high income quartiles, and the error bars indicate 95% confidence intervals.

Supplementary Fig. 26 | Associations between travel diversity and experienced segregation across low-income neighbourhoods with and without public transit stops within a 15-minute walking distance. a-d. Experienced segregation by travel diversity between low-income neighbourhoods with and without 15-minute public transits across urbanicity levels. **e-h.** The coefficients of travel diversity for each group across urbanicity levels from multiple linear regression models ($n = 83, 879$ CBGs for metropolitan areas, $n = 10,279$ for micropolitan areas, $n = 5,413$ for small towns, and $n = 4,337$ for rural areas). The dark colours indicate high income quartiles, and the error bars indicate 95% confidence intervals.

9. For Figure 1a, please provide the range of the experienced segregation index for each category, e.g., Low (value ~ value), etc.

Response: Thank you for this good suggestion. We have added the range of experienced segregation for each category, please see the response below.

10. For Figures 1b, 1c, and 1d, please provide a metropolitan-area-wide experienced segregation index for the Los Angeles, Chicago, and Houston metropolitan areas.

Response: Thank you for this suggestion. We re-mapped the figure by replacing the original examples of Los Angeles, Chicago, and Houston with the metropolitan statistical areas (MSAs) of Miami, Los Angeles, and Boston. This change was made because the average experienced segregation levels in Los Angeles, Chicago, and Houston were found to be relatively similar. To ensure the representativeness of the results, we selected three MSAs that exhibit distinct levels of segregation. We also added MSA-wide averages of experienced segregation for these three regions, as shown below (Line 149-160):

Fig. 1 | Patterns of experienced segregation. **a.** Spatial patterns of experienced segregation in the contiguous United States (presented here at the census tract level rather than the CBG level for the purpose of data visualization). We selected and zoomed in on three representative metropolitan statistical areas (MSAs) with distinct levels of experienced segregation: **b.** Miami–Fort Lauderdale–Pompano Beach (average segregation: 0.559), **c.** Los Angeles–Long Beach–Anaheim (average segregation: 0.602), and **d.** Boston–Cambridge–Newton (average segregation: 0.651). The experienced segregation was categorized into quintiles, with quintiles 1 to 5 corresponding to low, moderate-low, moderate, moderate-high, and high levels, respectively. **e.** Distribution of overall experienced segregation across different urbanicity levels (Metropolitan areas: $n = 167,924$ CBGs, Micropolitan areas: $n = 20,633$ CBGs, Small towns: $n = 10,870$ CBGs, and Rural areas: $n = 8,829$ CBGs). The box plots present the 10th, 25th, 50th, 75th, and 90th percentiles. **f.** Distribution of overall experienced segregation across various income quartiles in distinct urbanicity levels, with income quartiles defined separately for each urbanicity level.

Reviewer #2 (Remarks to the Author):

Thank for the opportunity to review the paper, Varying relationships between travel behaviour and socioeconomic segregation across income groups and urbanicity “. This is an important paper can potentially make a significant contribution to understanding the intersection of travel behavior and socioeconomic segregation based on the study of different income groups and urban settings in the US. Using a vast dataset of 1.2 billion mobility records, the study estimates individuals' experienced socioeconomic segregation and examines how travel distance and diversity impact segregation levels among various income groups and urbanicity categories. The study's key finding is that longer travel distances and more diverse destinations reduce segregation for residents of low-income communities, particularly in less urbanized areas, while the effect is less evident or even opposite for higher-income groups.

Thank you for your detailed review and positive comments. According to your valuable comments and insightful suggestions, we have carefully revised our manuscript to better present the introduction, methods, results, and discussion.

1.The text is too technical, using terms like "CBGs" instead of "neighborhood" and "POI" instead of "activity site." It would be more effective to use conceptually relevant terms and clarify how they are technically measured (e.g., by CBGs or POIs) in this specific study.

Response: Thank you for this insightful suggestion. We agree that using conceptually meaningful terms enhances clarity. We have replaced “CBGs” with “neighbourhoods” and “POIs” with “activity sites” throughout the manuscript. We have also clarified in the nutshell of the methods how these conceptual terms are measured in our study, as shown below (Line 103-108).

“We estimated experienced segregation using a large-scale human mobility dataset comprising 1,226,450,275 origin–destination (OD) flows from census block groups (CBGs) to activity sites, defined by Points of Interest (POIs). ... We conducted our analyses at the CBG level, which we refer to as neighbourhood in this study.”

2. Both conceptually and analytically, the paper does not adequately differentiate between residential segregation and non-residential segregation when moving around. This distinction is critical given the study's objective of problematizing the 15-Minute City concept and its underlying focus on promoting localized living. While the 15-Minute City concept may not be problematic in residentially mixed cities, it could become highly problematic in cities with pronounced residential segregation. Since neighborhood income context data is available, the study could examine the relationship between residential and non-residential segregation across different urban contexts and travel distances.

Response: Thank you for this valuable and important suggestion. In response, we additionally calculated residential segregation and examined its relationships with experienced segregation across social groups and urbanicity levels. Given that we measured experienced segregation as the income dissimilarity between a focal neighbourhood and the neighbourhoods of visitors present at the same activity sites, we adopted a parallel approach to measure residential

segregation. Specifically, residential segregation was calculated as the income dissimilarity between a given neighbourhood and its adjacent neighbourhoods. This approach allows for a consistent comparison between experienced and residential segregation, as detailed below (Line 674-683):

Measuring residential income segregation

We further measured residential segregation using the same income dissimilarity matrix to examine its relationship with experienced segregation. We defined overall residential income dissimilarity for each neighbourhood as the population-weighted mean income dissimilarity between the focal neighbourhood and its adjacent neighbours. The adjacent neighbour was determined using the queen contiguity method, which considers two neighbourhoods adjacent if their boundaries share either a common edge or vertex. The residential segregation of CBG_i was then calculated as:

$$S_{i,residential} = 1 - \frac{\sum_j P_j \times D_{i \rightarrow j}}{\sum_j P_j}$$

Where P_j is the population of the adjacent neighbourhood CBG_j relative to CBG_i

We have added results of the relationships between experienced segregation and residential segregation accordingly in the Results section, detailed as below (Line 162-215):

Comparison of experienced and residential segregation

To further examine the relationship between experienced and residential segregation, we calculated residential segregation for each neighbourhood. Similar to experienced segregation, residential segregation was defined as the weighted average income dissimilarity between a given neighbourhood and its adjacent neighbours, with weights based on the population size of each neighbour. Neighbourhood adjacency was determined using the queen contiguity approach, which has been widely used in spatial analysis. Unlike experienced segregation, residential segregation is relatively higher in both metropolitan and rural areas. In metropolitan areas, neighbourhoods in both the lowest and highest income quartiles exhibit elevated levels of residential segregation, which we term as ‘self-segregation’. In contrast, a clear income gradient in residential segregation is observed in less urbanized areas, with segregation levels decreasing as income increases (**Supplementary Fig. 5**). The distribution of experienced segregation closely mirrors that of residential segregation across urbanicity levels (Spearman correlation: 0.64 in metropolitan areas, 0.61 in micropolitan areas, 0.72 in small towns, and 0.70 in rural areas; $p < 0.001$), with experienced segregation generally lower. Specifically, more than 70% of neighbourhoods exhibit lower experienced segregation than residential segregation across all urbanicity levels: 78.2% in metropolitan areas, 72.4% in micropolitan areas, 71.7% in small towns, and 78.5% in rural areas (**Fig. 2a**). Among these neighbourhoods, the average reduction in experienced relative to residential segregation is greatest in metropolitan areas (18.19%, 95% CI: 18.13–18.24), and smaller in small towns (12.51%, 95% CI: 12.35–12.68) and rural areas (13.24%, 95% CI: 13.07–13.41) (**Fig. 2b**).

Regarding income disparities, neighbourhoods in the lowest income quartile in metropolitan areas exhibit the greatest reduction in segregation (23.32%, 95% CI: 23.21–23.43) when comparing experienced segregation to residential segregation, followed by those in the 2nd quartile (18.66%, 95% CI: 18.54–18.77), the 3rd quartile (15.41%, 95% CI: 15.31–15.51), and the highest-income neighbourhoods, which showed the smallest reduction (14.89%, 95% CI: 14.81–14.97). A similar pattern is observed in micropolitan areas and small towns, where neighbourhoods in the lowest income quartile also experience the largest reduction in segregation, while this trend is less clear in rural areas (**Fig. 2c**). To facilitate comparison, we grouped neighbourhoods in the 1st and 2nd income quartiles as low-income group, and those in the 3rd and 4th quartiles as high-income group. For racial disparities, non-white majority neighbourhoods, particularly those with low income, present greater segregation reduction across urbanicity levels (**Supplementary Fig. 6 and Supplementary Fig. 7**). This indicates that socially disadvantaged neighbourhoods (i.e., non-white majority neighbourhoods with low income) experience greater reduced income segregation in activity sites compared to residential areas, especially in metropolitan areas. The result shows that low-income neighbourhoods with relatively high residential segregation in highly urbanized areas experienced the greatest reduction in segregation, whereas high-income neighbourhoods with lower residential segregation across all urbanicity levels show the smallest reduction (**Fig. 2d**).

Fig. 2 | Relationships between experienced segregation and residential segregation. **a.** Scatter plots and Spearman correlations between experienced and residential segregation across urbanicity levels at the neighbourhood level. The grey dashed line represents the line of equality ($y = x$), indicating equal levels of experienced and residential segregation. The bar chart in the upper left corner shows the proportion of neighbourhoods with lower experienced segregation than residential segregation. **b.** Reduction in experienced segregation relative to residential segregation across urbanicity levels, calculated as $(\text{residential segregation} - \text{experienced segregation}) / \text{residential segregation}$. Only neighbourhoods where experienced segregation is lower than residential segregation were included in the calculation. **c.** Income disparities in segregation reduction across urbanicity levels. Error bars indicate 95% confidence intervals. **d.** Segregation reduction by income and residential segregation levels. In each urbanicity level, neighbourhoods in the 1st and 2nd income quartiles were classified as low-income group, and those in the 3rd and 4th quartiles as high-income group. Within each income group of each urbanicity level, neighbourhoods were further divided into low and high residential segregation groups based on the group-specific median. Error bars represent 95% confidence intervals.

Supplementary Fig. 5 | Distribution of residential segregation across urbanicity levels and income groups. **a.** Distribution of residential segregation across different urbanicity levels **b.** Distribution of residential segregation across various income quartiles in distinct urbanicity levels, with income quartiles defined separately for each urbanicity level.

Supplementary Fig. 6 | Segregation reduction between non-white and white majority neighbourhoods across urbanicity levels. Reduction in experienced segregation relative to residential segregation across urbanicity levels were calculated as $(\text{residential segregation} - \text{experienced segregation}) / \text{residential segregation}$.

Supplementary Fig. 7 | Segregation reduction between across income-racial groups. Reduction in experienced segregation relative to residential segregation across urbanicity levels were calculated as $(\text{residential segregation} - \text{experienced segregation}) / \text{residential segregation}$.

Moreover, we also compared experienced segregation during trips of varying travel distances to residential segregation, as shown below (Line 253-264):

We further compared experienced segregation during trips of varying travel distances to residential segregation and observed a clear distance gradient: longer trips to activity sites are associated with greater reductions in segregation relative to residential segregation. This gradient is particularly steep in rural areas, where the reduction in experienced segregation compared to residential segregation during the longest trips is 88.14% greater than that observed during the shortest trips. In contrast, in metropolitan areas, the corresponding difference is 53.40% (Fig. 3f). These results suggest that longer-distance activities more effectively mitigate residential segregation than shorter-distance activities, particularly in less urbanized settings. With respect to income disparities, low-income neighbourhoods, especially those with higher residential segregation, exhibit substantially greater segregation reductions during the longest trips compared to the shortest, most notably in less urbanized areas (Fig. 3g–j, Supplementary Fig. 15).

Fig. 3 | Patterns of experienced segregation decomposed by travel distance across urbanicity levels. a. Decomposition of segregation by trip distance quartiles (from the 1st to 4th quartile) across urbanicity levels. **b-e.** Evident income disparities in experienced segregation by travel distance quartiles across urbanicity levels. The box plots present the 10th, 25th, 50th, 75th, and 90th percentiles. **f.** Reduction in experienced segregation relative to residential segregation by travel distance quartiles across urbanicity levels. Only neighbourhoods where overall experienced segregation is lower than residential segregation were included in the calculation. **g-h.** Income disparities in segregation reduction by travel distance quartiles across urbanicity levels. Error bars indicate 95% confidence intervals.

Supplementary Fig. 15 | Reduction in experienced segregation relative to residential segregation by travel distance quartiles across income and residential levels and urbanicity levels. Noted that Only neighbourhoods where overall experienced segregation is lower than residential segregation were included in the calculation.

3. The study does not incorporate well the key concepts "activity space" and does not relate to more recent conceptual advancements such as "cocooning," "multicontextual segregation," "domains of segregation," or "vicious circles of segregation." POI is a technical term, but "activity site" is the underlying conceptual construct running through all these frameworks, but is missing in this study. Like-wise, "co-presence" and "co-location" should be conceptualised.

Response: Thank you for your valuable suggestion. As noted in our previous response, we have replaced "POIs" with "activity sites" throughout the manuscript. We have also restructured the Introduction to more clearly connect the concept of activity space to recent literature, elucidate the distinction between experienced and residential segregation (Line 24-29), and discuss the

role of activity sites under the context of localized living models, as shown below (Line 42-46).

“However, individuals also experience income segregation across a range of activity sites beyond their residential areas. Recent studies suggest that experienced income segregation (hereafter ‘experienced segregation’) in these non-residential settings is generally lower than that observed in residential areas¹¹. This multicontextual perspective provides important theoretical framework to understand how income segregation is shaped by residents’ daily travel and activities.” ...

... “Moreover, activity sites within localized living circles may be predominantly occupied by socially homogeneous populations from surrounding neighbourhoods, particularly in areas with high residential segregation, thereby intensifying segregation by limiting opportunities to access activity sites beyond the residential environment²¹.”

To enhance conceptual clarity, we defined “co-location” as the presence of individuals in the same activity site, as shown below (Line 115-118).

“Experienced segregation of individuals from a neighbourhood in each activity site was calculated as the weighted average income dissimilarity between their neighbourhood and the neighbourhoods of other individuals present in the same space, with weights based on the number of individuals from each other neighbourhood.”

4. Not all activity sites and related activities have the same significance in terms of experienced segregation. For example, schools and cinemas have different dynamics of social interaction. It would be beneficial to relate the study to activity-based research to capture these distinctions more effectively.

Response: Thank you for highlighting this important point. We agree that the intensity of social interactions varies across different types of activity sites, and thus activity sites contribute unequally to experienced segregation. However, due to limitations in the mobility data, distinguishing interaction intensity across sites remains a methodological challenge and is yet to be fully addressed in existing research. Similar to prior studies, we measure experienced segregation using the likelihood of co-location—defined as the probability that individuals from different neighbourhoods are present at the same activity site on a weekly basis—as a proxy for social exposure. To assess the robustness of this measure, we conducted a sensitivity analysis by recalculating experienced segregation using only selected activity sites that, based on prior literature, are assumed to involve higher levels of social interaction, while excluding those likely to involve limited interaction. The resulting measure was highly correlated with our original estimates, suggesting minimal bias and supporting the robustness of our findings (Line 661-671).

... “Notably, not all activity sites contribute equally to social interaction and, by extension, to experienced segregation. For instance, interactions may be limited in some settings, e.g., theatres and cinemas, but more frequent in socially intensive environments such as schools. Although our primary approach does not differentiate activity sites by their propensity to facilitate interaction, we conducted another sensitivity analysis by recalculating experienced

segregation using a subset of eight activity site types identified as more conducive to social interaction, including groceries, markets, parks, religious venues, restaurants, schools, sport facilities, and shopping stores^{57,58}. The resulting measure $S_{i,selected}$ shows an almost perfect correlation with the original index S_i (spearman's $R = 1.00$, $p < 0.001$), suggesting that the influence of excluding this differentiation on the overall estimates is negligible (**Supplementary Fig. 29**).”

Supplementary Fig. 29 | Scatter plot and Spearman correlation between the main measure of overall experienced segregation and the alternative measure based on selected types of activity sites. Each point represents a neighbourhood, with the x-axis showing experienced segregation from the main analysis and the y-axis showing the adjusted measure that includes only activity sites assumed to involve higher levels of social interaction. The high correlation indicates the robustness of the main measure.

In addition, we have acknowledged this limitation in the Discussion section and offered suggestions for future research (Line 499-513).

“Finally, it should be noted that different types of activity sites vary in their potential to facilitate social interaction and thus may contribute unequally to experienced segregation^{11,52}. For instance, individuals are likely to engage in markedly different levels of interaction in interaction-restricted environments (e.g., theatres and cinemas) compared to interaction-intensive settings (e.g., schools and parks), even with equal visiting duration. Although this distinction is critical, it remains challenging to quantify due to the absence of interaction-specific information in the mobility data or previous research. We conducted a sensitivity

analysis by recalculating experienced segregation using visits to eight selected types of activity sites identified in prior research as particularly conducive to social interaction. The results are highly consistent with those calculated using the full set of activity spaces, suggesting that the omission of interaction intensity across space types has minimal impact on the overall measurement of experienced segregation. Future studies are encouraged to integrate mobility data with survey-based measures of social interaction intensity across different activity spaces, to improve the accuracy of experienced segregation estimates.”

5. The study lacks clarity on the type of contextual information available about activity sites, such as schools. How exactly does the study capture the residential context, the context of other activity sites, the spatial resolution, and the duration of stay in these sites? This information is necessary to understand the study’s contribution comprehensively.

Response: Thank you for your question. In terms of the residential context, the SafeGraph dataset identifies the home location of each device by analysing nighttime location data (between 6 p.m. and 7 a.m.) over a 6-week period. These home locations are assigned at the Geohash-7 level and subsequently mapped to census block groups (CBGs). We have clarified this procedure in the Methods section, as shown below (Line 524-528):

“Home locations are determined by SafeGraph using nighttime data (6 pm to 7 am) over a 6-week period, with each device’s location assigned at the geodash-7 level and mapped to CBGs. Hence, in this study, home CBG identifiers in the mobility records were used to represent the residential neighbourhoods of visitors within each OD flow.”

Regarding the context of activity sites, we have supplemented the Methods section with a detailed description of the types of activity sites included in this study, as shown below (Line 533-541):

“Each POI was classified using six-digit North American Industry Classification System (NAICS) codes from SafeGraph’s Core Places dataset. Our analyses focused on 16 representative types of activity sites, which together account for over 70% of all locations in the dataset and reflect a broad range of resident needs: basic living (markets, restaurants, life services, and hotels); health and well-being (healthcare facilities, hospitals, and personal care); cultural and spiritual engagement (cultural and religious facilities); education and social development (schools and social assistance); recreation and leisure (entertainment venues, sports facilities, and parks); and consumer and retail activities (shopping stores and groceries).”

With respect to spatial and temporal resolution, the SafeGraph dataset provides aggregated mobility data at the CBG level (neighbourhoods) on a weekly basis throughout 2019. We have added a discussion of how these spatial and temporal resolutions may influence our analyses in the Limitations section, as detailed below (Line 488-499):

“First, as mobility records are aggregated at the CBG level, we conducted analyses at the neighbourhood level rather than the individual level. However, CBGs are the smallest spatial unit with available both mobility data and socioeconomic data, providing sufficient spatial resolution and a large representative sample for neighbourhood-scale analyses. Second, given

the temporal resolution of the dataset, our analyses are limited to estimating the likelihood of co-location, namely the probability that individuals from different neighbourhoods are present at the same activity site, on a weekly basis, which serves as a proxy for social exposure. However, individuals from a neighbourhood may visit the same place at different times or on different days of the week. While our approach does not capture more precise temporal overlap, weekly mobility records provide a reliable approximation of individuals' general travel patterns. Accordingly, the experienced exposure measured in this study are likely reflect real-world exposure with reliable accuracy.”

For the duration of stay at activity sites, the SafeGraph dataset provides only the median dwell time for POI, without distinguishing dwell times by visitors' neighbourhoods. Consequently, our main analyses did not incorporate dwell time in measuring experienced segregation. To address this limitation, we conducted a sensitivity analysis by recalculating experienced segregation weighted by the median dwell time of each activity site, assuming equal dwell time for visitors from all neighbourhoods within the same site. The resulting measure showed a high correlation with our original calculation, demonstrating the robustness of our findings. We have incorporated this information in the Methods section, as detailed below (Line 649-661):

“To assess the robustness of our approach, we re-performed the calculation by incorporating the median dwell time for each activity site, thereby accounting for variation in the intensity of exposure across different activity site. Specifically, we recalculated the overall experienced income dissimilarity for CBG_i in week w by weighting experienced income dissimilarity at each activity site by the product of the number of visitors and the median dwell time at POI n . The experienced segregation for CBG_i over the entire year was then re-calculated as:

$$S'_i = 1 - \frac{\sum_w \sum_n (V_{i,n,w} \cdot T_{n,w}) \times E_{(k_i,n,w)}}{\sum_w \sum_n (V_{i,n,w} \cdot T_{n,w})}$$

where $T_{n,w}$ is the median dwell time for POI n in week w . S'_i is highly correlated with S_i (Spearman's $R = 0.99$, $p < 0.001$), indicating strong consistency and robustness of our results (**Supplementary Fig. 27**). The high correlation between two indices is consistent across all type of activity sites, with Spearman's R ranging from 0.84 in life service settings ($p < 0.001$) to 1.00 in hospitals ($p < 0.001$) (**Supplementary Fig. 28**).”

Supplementary Fig. 27 | Scatter plot and Spearman correlation between the main measure of overall experienced segregation and the alternative measure incorporating dwell time. Each point represents a neighbourhood, with the x-axis showing experienced segregation from the main analysis and the y-axis showing the adjusted measure accounting for median dwell time at activity sites. The high correlation indicates the robustness of the main measure.

Supplementary Fig. 28 | Scatter plot and Spearman correlation between the main measure of overall experienced segregation and the alternative measure incorporating dwell time by activity sites. Each point represents a neighbourhood, with the x-axis showing experienced segregation from the main analysis and the y-axis showing the adjusted measure accounting for median dwell time at activity sites. The high correlation indicates the robustness of the main measure.

6. Although methodological details are provided in a separate section, a nutshell information should be available earlier in the manuscript. For instance, the first paragraph in the "Results" section states that "the overall median experienced segregation across all CBGs was 0.594." Without any context, this statement does not carry meaning. It is necessary to clarify, in nutshell: (a) what CBGs represent, (b) the groups being studied, (c) what is experienced segregation, and (d) how what is the measure used, including its meaning and range.

Response: Thank you for this valuable suggestion. We agree that providing a brief summary of the methods prior to the results would enhance clarity and aid reader understanding.

Accordingly, we have added a concise overview of the methods used in this study in the opening paragraph of the Results section, as shown below (Line 103-125).

“We estimated experienced segregation using a large-scale human mobility dataset comprising 1,226,450,275 origin–destination (OD) flows from census block groups (CBGs) to activity sites, defined by Points of Interest (POIs). Even though CBGs are the next level above census blocks, income data from ACS are not available at the block level, CBGs are thus the smallest geographic units for which mobility and income data are available. We conducted our analyses at the CBG level, which we refer to as neighbourhood in this study. To quantify experienced segregation for residents of each neighbourhood, we first constructed an income dissimilarity matrix using the ranked median household income of each neighbourhood, following the concept of ‘income distance’ developed by Xu et al.³⁴ The income dissimilarity from a focal neighbourhood to another is defined as the proportion of all other neighbourhoods whose income distance to the focal neighbourhood is smaller than that between the two (see Methods for details). This matrix captures the income dissimilarity between all pairs of neighbourhoods. Experienced segregation of individuals from a neighbourhood in each activity site was calculated as the weighted average income dissimilarity between their neighbourhood and the neighbourhoods of other individuals present in the same space, with weights based on the number of individuals from each other neighbourhood. The overall experienced segregation of each neighbourhood was calculated as the weighted average of segregation across all activity sites visited by its residents, with weights based on the number of individuals from the given neighbourhood present in each space. Values of experienced segregation range from 0 to 1, with higher values indicating greater levels of segregation. To determine the heterogeneity of experienced segregation across urbanicity, we categorized neighbourhoods into four urbanicity levels based on the Rural–Urban Commuting Area (RUCA) Codes: metropolitan areas, micropolitan areas, small towns, and rural areas.”

7. The finding that experienced segregation decreases with increasing metropolitan size is unexpected and contradicts much of the existing literature on residential segregation by city size. This result requires more extensive discussion, linking to previous studies and exploring possible explanations. May it be due to mobility patterns, increased access to diverse activity sites, or other factors?

Response: Thank you for this comment. We additionally examined residential segregation across levels of urbanicity and found that it is relatively high in both metropolitan and rural areas, which is different from experienced segregation. In metropolitan regions, neighbourhoods in both the lowest and highest income quartiles exhibit elevated residential segregation. We have added a detailed discussion to illustrate this distinction between residential and experienced segregation, as show below (Line 409-421):

“First, our study highlights how experienced segregation differs from residential segregation across social groups and urbanicity levels. While experienced segregation is strongly correlated with residential segregation, it tends to be lower, particularly in metropolitan areas. While a clear gradient of decreasing experienced segregation is observed with increasing urbanicity, residential segregation remains high in metropolitan settings, where self-segregation is evident

among both the lowest- and highest-income neighbourhoods. Prior research has documented the prevalence of income inequality and self-segregation under residential contexts in highly urbanized areas, contributing to increased levels of residential income segregation in these areas³⁸⁻⁴⁰. Our study builds upon this understanding by revealing that more urbanized environments, characterised by greater land-use diversity and a greater variety of amenities, may offer increased opportunities for cross-group interactions in activity sites^{41,42}. These environments foster social integration by enabling encounters among individuals from diverse socioeconomic backgrounds beyond residential areas.”

8. The results on trip length and segregation are important. However, it would be valuable to further explore whether longer trips specifically reduce experienced segregation in highly segregated cities, potentially revealing nuanced dynamics of residential and non-residential segregation.

Response: Thank you for this insightful suggestion. In response, we conducted additional analyses as recommended. The results match your prediction. Specifically, we grouped neighbourhoods in the 1st and 2nd income quartiles as low-income group, and those in the 3rd and 4th quartiles as high-income group. Within each income group, we further classified neighbourhoods into high and low residential segregation based on the median segregation value. Building on this classification, we compared the relationships between experienced segregation and travel distance using both intra-group and inter-group analysis. The details of these analyses are provided below:

1) Experienced segregation by trips of varying travel distances (Line 218-225, Line 247-249)

“To compare experienced segregation by trips with different travel distance, we first classified all trips as either inside-neighbourhood or outside-neighbourhood for each neighbourhood, assuming that trips beyond the home neighbourhood involve greater travel distances. ... Particularly, both high- and low-income neighbourhoods with high residential segregation demonstrate greater reductions in experienced segregation during trips outside their own neighbourhoods relative to trips inside them (**Supplementary Fig. 9**).”

Supplementary Fig. 9 | Experienced segregation for inside-neighbourhood and outside-neighbourhood trips by income and residential segregation levels across urbanicity levels.

... “In addition, neighbourhoods with high residential segregation, especially those with low-income, show greater reduction in experienced segregation during the longest trips compared to the shortest ones (Supplementary Fig. 12).” ...

Supplementary Fig. 12 | Patterns of experienced segregation decomposed by travel distance across neighbourhoods with different income and residential segregation levels and urbanicity levels.

2) Neighbourhood travel behaviour and experienced segregation (Line 365-373)

“To further investigate heterogeneity in the observed associations, we conducted subsample analyses exclusively for low-income neighbourhoods (i.e., those in the 1st and 2nd income quartiles), examining whether the negative relationships between travel distance, travel diversity, and experienced segregation differ across neighbourhoods with varying levels of residential segregation, racial majorities, and access to public transit. Among low-income neighbourhoods in metropolitan areas, small towns and rural areas, those with higher residential

segregation present stronger negative relationships between experienced segregation and mean travel distance, indicating that these neighbourhoods with longer mean travel distances tend to experience greater reduction in segregation (Supplementary Fig. 21).”

Supplementary Fig. 21 | Associations between mean travel distance and experienced segregation between low-income neighbourhoods with high and low residential segregation. a-d. Experienced segregation by mean travel distance between low-income neighbourhoods with high and low residential segregation across urbanicity levels. **e-h.** The coefficients of travel distance for each group across urbanicity levels from multiple linear regression models ($n = 83, 879$ CBGs for metropolitan areas, $n = 10,279$ for micropolitan areas, $n = 5,413$ for small towns, and $n = 4,337$ for rural areas). The dark colours indicate high income quartiles, and the error bars indicate 95% confidence intervals.

We also added a detailed discussion of these findings in the Discussion section, highlighting how the relationship between experienced segregation and travel distance varies across income and residential segregation groups, and offering implications for these patterns (Line 427-438).

“Second, our study demonstrates that experienced segregation is linked to travel distance from a home neighbourhood, based on intra-neighbourhood analyses. Travelling to activity sites over longer distances from one’s home neighbourhood are more likely to reduce experienced segregation and help mitigate residential segregation than travelling to those over shorter distance. This effect is especially pronounced among neighbourhoods with high levels of residential segregation, suggesting that long-distance travel may offer greater benefits of reduced income segregation for these neighbourhoods. These findings challenge the localized living models, which may work well in socially mixed neighbourhoods, where cross-class interactions are feasible even when their residents’ activities are largely confined to a limited geographic area. However, in neighbourhoods characterized by high residential segregation, localized living within a socially homogeneous environment risks reinforcing segregation by restricting residents’ exposure to diverse social contexts.”

9. The Discussion section should link back to the 15-Minute City concept and localized living in different urban contexts. Also, the discussion of the specific context of the US with large sprawled city-regions and automobile-based travel should be addressed to increase the international impact. These extensions could be made by significantly cutting back and streamlining existing discussions.

Response: Thank you for this insightful suggestion. We strongly agree that linking the concept of localized living models to our findings strengthens the discussion and deepens the understanding of how such models may inadvertently reinforce segregation. We also recognize the importance of situating our analysis within the specific context of the United States to enhance the international impact of our study. Accordingly, we have restructured the Discussion section to include a detailed demonstration of our results under the concept of localized living models, and we have expanded the discussion to reflect the specific context of the United States, please see more details in the revised manuscript (Line 404-487).

Reviewer #1 (Remarks to the Author):

I would like to thank the authors for addressing my previous comments. I have the following remaining – hopefully minor – comments, which I hope the authors could also address:

1. Regarding my previous comment #7 (subsample analysis by racial profile of communities): as shown in Supplementary Figures 3-4 in this revision, the levels of experienced segregation between white-majority and non-white-majority communities differ between metropolitan areas and the other three settlement types (i.e., micropolitan, small town, and rural). What could explain this pattern? Could it be due to differences in trip distance, or something else? Some discussions on this would enhance the paper.

Response: Thank you for this valuable suggestion. We have added several possible explanations for patterns presented in Supplementary Fig. 3-4, which are shown as below:

...The relatively high accessibility to public transit and more transportation options in metropolitan areas enables residents to travel more easily to various destinations and interact with people from diverse socioeconomic backgrounds compared to those in less urbanized regions³⁵. Majority-POC neighbourhoods, particularly those with low income, are likely to benefit more from these enhanced opportunities for mobility and social interaction in metropolitan settings. In contrast, in less urbanized areas, limited accessibility to amenities constrains residents of low-income, majority-POC neighbourhoods from visiting areas frequented by more diverse populations. Meanwhile, residents of high-income neighbourhoods, regardless of racial composition, generally have greater access to resources and mobility, allowing them more opportunities to visit destinations with heterogeneous visitors.

Reference: Wang, W., Espeland, S., Barajas, J. M. & Rowangould, D. Rural–nonrural divide in car access and unmet travel need in the United States. *Transportation* 52, 507–536 (2025).

2. For my previous comment #9 (range of experienced segregation index in the map): in the revised Figure 1a, the legend is currently written as follows—low (≤ 0.529), moderate low (≤ 0.570), etc. I believe this should be corrected for clarity and accuracy. For example: low (≤ 0.529), moderate low (0.530~0.570), and so on. Please update this accordingly.

Response: Thank you for your suggestion. We have revised the legend of Figure 1a to clearly indicate the exact range of values for each category to enhance clarity and accuracy. The updated map is shown below:

3. I am not sure whether the term “segregation reduction”, as used in Supplementary Figure 6 and elsewhere in the manuscript, is appropriate. The term "reduction" typically implies temporal change or causal inference, which does not appear to be the case in this context. Perhaps a term like "residential-experience segregation disparity" or something similar would be more suitable.

Response: Thank you for highlighting this important point. We agree that the term “reduction” may imply temporal change or causal inference. Accordingly, we have revised the terminology to “residential–experienced segregation disparity” in all figures and updated the corresponding expressions throughout the manuscript for greater precision and clarity.

Reviewer #2 (Remarks to the Author):

Thank you very much, the response letter is very well organized, and I am happy with the thorough and thoughtful revisions.

Response: Thank you for your positive feedback. We sincerely appreciate your valuable comments throughout this process and hope that our work will contribute meaningfully to the field.